# Towards Lightweight Black-Box Attacks Against Deep Neural Networks

**Chenghao Sun**[1]  **Yonggang Zhang**[2]  **Wan Chaoqun**[3]  **Qizhou Wang**[2]
**Ya Li**[4]  **Tongliang Liu**[5]  **Bo Han**[2]  **Xinmei Tian**[1*]
[1]University of Science and Technology of China  [2]Hong Kong Baptist University
[3]Alibaba Cloud Computing Ltd  [4]iFlytek Research
[5]The University of Sydney

## Abstract

Black-box attacks can generate adversarial examples without accessing the parameters of deep neural networks (DNNs), largely exacerbating the threats of deployed models. However, previous works state that black-box attacks fail to mislead DNNs when their training data and outputs are inaccessible. In this work, we argue that black-box attacks can pose practical attacks in this highly restrictive scenario where only several test samples are available. Specifically, we find that attacking the shallow layers of DNNs trained on a few test samples can generate powerful adversarial examples. As only a few samples are required, we refer to these attacks as *lightweight* black-box attacks. The main challenge to promoting lightweight attacks is to mitigate the adverse impact caused by the approximation error of shallow layers. As it is hard to mitigate the approximation error with few available samples, we propose Error TransFormer (ETF) for lightweight attacks. Namely, ETF transforms the approximation error in the parameter space into a perturbation in the feature space and alleviates the error by disturbing features. In our experiments, lightweight black-box attacks with the proposed ETF achieve surprising results. For example, even if only 1 sample per category is available, the attack success rate achieved by lightweight black-box attacks is only about $3\%$ lower than that of the black-box attacks using complete training data [1].

## 1 Introduction

Black-box attack methods [12, 56, 60] can mount successful attacks without accessing the parameters of deep neural networks (DNNs), posing great challenges to deep learning in safety-critical situations. Existing black-box attack methods implicitly assume that the training data and/or the outputs of target models are available to adversaries. For scenarios where training data are available, adversarial examples can be generated by attacking surrogate models constructed for approximate target models. Using the target models' outputs for estimating gradients is another practical approach to crafting adversarial examples. In general, due to the low dependency on target model information, the black-box attacks raise realistic threats to real-world deep learning systems, including semantic segmentation [2, 36], object detection [18, 45, 44], and automatic driving [54].

However, these assumptions can be violated in many practical scenarios [3, 46], where both the training data and the outputs of target models are inaccessible. This realistic attack scenario is introduced in [3], generally termed as the *no-box setting*. Under such a restrictive setting, Li et

---

[*]Corresponding author (xinmei@ustc.edu.cn)

[1]Code is available at `https://github.com/sunch-ustc/Error_TransFormer/tree/ETF`

al. [34] stated that existing black-box attacks fail to mislead target models because it is now forbidden to access previously essential information in mounting black-box attacks. Therein, the failure of existing black-box attacks is not surprising. The reason is that constructing reliable surrogate models typically relies on accessing the complete training data of target models, which is forbidden in such a restrictive setting with only a few test samples available.

In this work, we aim to reveal the potential threats of black-box attacks in the no-box setting, as mounting black-box attacks with highly limited accessible information leaves many models exposed to attack. Specifically, we challenge the previous believes by raising the following question: *Can black-box attacks success in the no-box setting?* The doubts about the potential threats of the black-box attack are not groundless. Specifically, the potential threats of black-box attacks stem from two facts: a) adversarial examples can be generated by perturbing representations at shallow layers of DNNs [28, 27, 48]; b) regarding the representation of shallow layers, there do not exist critical differences between those models learned from a few data and that of the whole training data. [1]

Building upon the above facts, it is actually possible that an adversary can successfully mount attacks using limited samples, as powerful adversarial examples can be generated by attacking the shallow layers of DNNs trained on few samples. Concretely, the adversary can leverage available samples to construct a surrogate model to approximate the shallow layers of target models within acceptable errors. Consequently, adversarial examples can be generated by perturbing the features obtained from the shallow layers of the constructed surrogate model. As merely a few samples are required for attacking, we refer to black-box attacks in the no-box setting as *lightweight black-box attacks*. Intuitively, the closer the surrogate model is to the target model, the higher the attack success rate the crafted adversarial examples have [31]. If the approximation error of shallow layers is alleviated, the lightweight black-box attack can be as powerful as black-box attacks with complete training data. Hence, the main challenge to mounting a lightweight attack is to mitigate the adverse impact caused by the approximation error of shallow layers.

However, it is challenging to mitigate the approximation error, especially when the number of available samples is limited. Fortunately, it is straightforward to identify which kind of perturbations are preferred: if perturbations applied to a feature contribute to fooling the surrogate model, the perturbation is preferred. Therefore, bridging the connection between the parameter space and the feature space is the key to mitigating the approximation error. Accordingly, we propose transforming the approximation error in the parameter space as the perturbation in the feature space, dubbed Error TransFormer (ETF). Namely, ETF transforms the worst-case approximation error to the worst-case feature perturbation, leading to a min-max scheme that generates adversarial examples under the worst feature perturbations. We verify the attack success rate of lightweight black-box attacks using 7 models trained on the ImageNet dataset [47] and find that existing attack methods are much more potent than previously claimed. Moreover, the performance of lightweight black-box attacks can be further promoted by the proposed ETF, i.e., the attack success rate is only $3\%$ lower than that achieved by black-box attacks having complete training data of target models.

## 2 Related works

### 2.1 Adversarial attack

According to the amount of accessible information, existing adversarial attacks can be roughly divided into two categories: white-box attacks and black-box attacks. White-box attacks [19, 41, 15, 16] assume that the target model is transparent to adversaries, i.e., adversaries can access all information about the target model. Nevertheless, the assumption of transparent target models can be violated in many practical scenarios [11]. Hence, black-box attacks [14, 37, 64, 7] are proposed and applied to the scenario where relatively limited information about the target model is accessible, i.e., only a certain number of model queries or the training data of target tasks are available.

Among these black-box attacks, intermediate-level attacks [48, 28, 24] are widely explored to improve the adversarial transferability. The core idea of these attacks is based on the empirical observation that well-trained models' intermediate features are transferable [58]. This is consistent with the recent works showing that adversarial examples comprise spurious features having the transferable property [63, 6]. Hence, Inkawhich et al. [28] propose to perturb the feature space of neural networks to create more transferable adversarial examples. These methods provide an interesting empirical conclusion that disturbing the shallow layers of DNNs can also generate adversarial examples.

Existing black-box attacks assume that the large-scale training data of target models and the feedback from querying target models are accessible, but these assumptions can be violated in many scenarios, e.g., the no-box setting [3]. Li et al. [34] stated that existing black-box attacks cannot be successfully mounted because existing DNNs require large-scale training data for generalization. Given only small-size data available, obtaining a surrogate model with strong generalization is challenging. Hence, Li et al. propose replacing black-box attacks with their proposed no-box attack [34], where they train a classical auto-encoder model instead of the supervised classification model due to the constraint of a small-scale dataset. Concretely, they train 20 auto-encoders for each category and generate adversarial examples by attacking these auto-encoders, which is time- and computational-consuming.

Different from existing black-box attacks, lightweight black-box attacks aim to reveal the potential risk of black-box attacks under the no-box setting. Because one main challenge to perform lightweight black-box attack is to mitigate the adverse impact caused by the approximation error, we propose error transformer that is a min-max strategy transforming the approximation error in the weight space to the feature space. The min-max strategy is different from that introduced in [55], where min-max strategy is proposed to flatten the loss landscape in the weight space, see Appendix F.

## 2.2 Approximation of Shallow Networks

A recent study [58] confirms the intuition that shallow layers in DNNs can be seen as low-level feature extractors, provided that strong data augmentation is used [1]. Motivated by this observation, Asano et al. [1] design a method to explore the information in every layer of DNNs. Asano et al. [1] train a supervision model on ImageNet LSVRC-12, and a self-supervision model on a small-scale dataset. Then, they apply linear probes [61] to all intermediate layers of networks, where a linear classifier is trained on the top of pre-trained and fixed feature representations. In this way, they evaluate the quality of the representation learned at different depths of the networks. The results show that, given heavily synthetic transformations, the shallow layers of DNNs learned from a few images can approximate that of DNNs trained on millions of images. According to the intriguing empirical observation, it is possible to construct a surrogate model with limited samples, where its shallow layers are similar to that of the target models. Besides, it is known that the neural networks prefer to make the decision through the spurious feature [63] on which are focused by models to correlate to the true label. Since shallow layers based on the small-scale or large-scale data set can acquire the similar spurious feature [1], it is an effective way to leverage the shallow model to mount an attack in the spurious features when the number of the data is limited.

# 3 Preliminaries

## 3.1 No-Box Threat Model

The no-box setting [3] denotes a threat model where available information about the target model is extremely limited, making it challenging to generate adversarial examples. According to the description in [3], the no-box threat model can be defined as follows.

**Definition 1 (No-Box Threat Model)** *A threat model is called the no-box threat model if the adversary in this scenario is not allowed to access the target model's training data and outputs, so it only has some samples that can be correctly predicted with high probability by the target model.*

The definition is consistent with the no-box setting introduced in [3], where the target model's training data and outputs are inaccessible to the adversary. Moreover, the available samples used for generating adversarial examples are expected to be limited. This is because if the number of samples is similar to or even larger than the number of samples used for training target models, the adversary can leverage these samples to construct a surrogate model similar to target models, making the condition of inaccessible training data meaningless. It is intuitive that target models can correctly predict the labels of these available samples with high probability. For example, it is meaningless to perform attacks using samples misclassified by the target model.

## 3.2 Lightweight Black-Box Attack

**Definition 2 (Lightweight Black-Box Attack)** *An attack is called the lightweight black-box attack if an adversary aims to perform an attack in the context of a no-box threat model.*

The definition of the lightweight black-box attack shows that lightweight attack is a special kind of black-box attacks. The extremely limited accessible information distinguishes it from other black-box attacks. For example, query-efficient black-box attacks require the feedback information of target models [25, 26], while lightweight attacks do not. To our best knowledge, how to perform a lightweight black-box attack is still lacking in the literature. For example, Li et al. [34] claim that black-box attacks fail to fool target models when their training data and outputs are inaccessible.

# 4 Approach

This section gives a detailed description of how to perform lightweight black-box attacks and the proposed Error TransFormer (ETF) to alleviate to adverse impact caused by approximation error.

## 4.1 Lightweight Surrogate Model

Surrogate models are widely used in black-box attacks since adversarial examples are usually generated by attacking surrogate models when querying target models is forbidden. Since the lightweight black-box attack is a special kind of black-box attack, performing the attack also requires constructing surrogate models. We call surrogate models employed in lightweight black-box attacks the *lightweight surrogate model*, building upon the fact that the number of available samples in the no-box threat model is typically limited.

Similar to existing black-box attacks, we assume that the label information of samples used for generating adversarial examples are accessible. In this scenario, the lightweight surrogate model is trained for a classification task. Specifically, the lightweight model is realized by training a classification DNN equipped via conventional supervised learning, using cross-entropy loss $\ell(\cdot, \cdot)$:

$$\min_w \mathbb{E}_{(x,y)\sim\hat{\mathcal{D}}} \ell(f(x;w), y), \tag{1}$$

where $w$ stands for the parameters of the surrogate model $f$, and $(x, y)$ denotes the sample label pair of random variables sampled from the distribution of natural data $\hat{\mathcal{D}}$.

In black-box attacks, adversary only leverages the shallow layers of DNNs, leading to a straightforward approach to constructing lightweight surrogate models. Specifically, we can train the lightweight surrogate model in a contrastive manner, where the supervised information is no longer necessary:

$$\min_w \frac{1}{|\hat{D}|} \sum_{\mathbf{x}\in\hat{D}} -\log \frac{\sigma(f(\mathbf{x};w), f(\mathcal{T}(\mathbf{x});w))}{\sum_{\mathbf{x}'\sim\hat{D}\backslash\mathbf{x}} \sigma(f(\mathbf{x};w), f(\mathcal{T}(\mathbf{x}');w))/(|\hat{D}\backslash\mathbf{x}|-1)}, \tag{2}$$

where $\sigma$ is a specific similarity metric, e.g., cosine similarity, $\mathcal{T}$ denotes the data transformation operation, and $\hat{D}\backslash\mathbf{x}$ means that $\mathbf{x}$ is removed from the dataset $\hat{D}$. The contrastive strategy in Eq. 2 makes it possible to perform attacks when the label information is unavailable. Unless otherwise specified, we mainly uses Eq. 1 to train the surrogate models.

## 4.2 Feature Space Perturbation

Built upon the empirical observation [58], using limited samples can approximate the shallow layers of DNNs within acceptable errors. This suggests that deep layers other than shallow layers can cause large approximation error. Accordingly, merely the shallow layers of the learned lightweight surrogate models are used for generating adversarial examples, which is similar to the feature space attacks [48]. For brevity, we define the shallow layers as $\varphi$ and denote the feature as $\varphi(x;w)$.

In feature space attacks, guide images are usually required for generating adversarial examples [48]. A natural image used for generating adversarial example is usually called source image. Given a source image $x_s$, we generate an adversarial example $x_{adv}$ by perturbing $x_s$ such that its feature is similar to that of a guide image $x_g$, where the labels of these two images are different. Specifically, the adversarial examples are generated as follows:

$$x_{adv} = \arg\min_{\|x'-x\|_p\leq\epsilon} d(\varphi(x_g;w), \varphi(x';w)), \tag{3}$$

where $d$ stands for a specific distance, $\epsilon$ controls the strength of adversarial perturbation, and $x'$ denotes the perturbed version of the source image.

## 4.3 Error Transformer

Although applying feature space attacks to lightweight surrogate models can generate transferable adversarial examples, the approximation error will cause adverse impact to mounting attacks. The reason is that large dissimilarity between the surrogate models and the target models reduces the attack success rate. Therefore, the main challenge to mounting lightweight attacks is to mitigate the adverse impact caused by the approximation error. However, available samples are usually limited in the no-box threat model, making it hard to alleviate the approximation error.

To address the challenge, we propose transforming the parameter space's approximation error as the feature space's perturbation. Specifically, we seldom know which perturbations can point (from the surrogate model) to the target model, making it challenging to alleviate the approximation error in the weight space. In contrast, we have the prior that samples with different labels should have distinguishable representations. Thus, we can leverage such prior knowledge to select preferred perturbations in the feature space, i.e., we prefer perturbations that can make representations of samples with different labels indistinguishable. Similarly, it is also straightforward to define the "bad" perturbations, leading to the design of the min-max optimization to identify the "worst" model. Thus, connecting the parameter space and feature space makes it possible to mitigate the adverse impact caused by approximation error. To bridge the connection, we introduce the following key identity, which is also used in [43]:

$$\varphi(x; \{w^1 + w^1 A\} \cup \{w \backslash w^1\}) = g((w^1 + w^1 A)x; w \backslash w^1) = g(w^1(x + Ax); w \backslash w^1) = \varphi(x + Ax; w), \quad (4)$$

where $w^1$ stands for the first layer parameters of model $\varphi$, $A$ is a transformation matrix used for perturbing $w^1$, $\{w^1 + w^1 A\} \cup \{w \backslash w^1\}$ means that the first layer's parameters are perturbed, the other layers' parameters keep unchanged, and $g$ is the function parameterized with $w \backslash w^1$ used for processing the first layer's outputs. According to Eq. 4, we can transform a perturbation in the parameter space as the perturbation in the data space, i.e., $\varphi(x; \{w^1 + w^1 A\} \cup \{w \backslash w^1\}) = \varphi(x + Ax; w)$. Similarly, we can apply the transformation operation to the hidden layers, see Appendix A for more details.

Built upon the connection in Eq. 4, we can transform the first layer parameters' discrepancy between the target model and the surrogate model. Taking the first layer as an example, let $w_t$ be the parameters of the target model and $w_t^1$ be its first layer parameters, where the other parameters are the same as the surrogate model. Assume [2] that there exists a transformation matrix $A$ applied to the first layer parameters of the surrogate model such that $w_t^1 = w^1 + w^1 A$, we have:

$$\varphi(x; w_t^1 \cup \{w_t \backslash w_t^1\}) = \varphi(x; \{w^1 + w^1 A\} \cup \{w_t \backslash w_t^1\}) = \varphi(x + Ax; \{w^1\} \cup \{w_t \backslash w_t^1\}) = \varphi(x + Ax; w).$$
$$(5)$$

The identity in Eq. 5 shows that we can find a perturbation in the data space to alleviate the difference in the first layer parameters between these two models. Built upon the transformation, we can transform the worst approximation error to the worst perturbation in the data space. Therefore, we can employ a min-max strategy to mitigate the adverse impact of approximation error through generating adversarial examples under the worst perturbation. Thus, lightweight black-box attacks with ETF generate adversarial examples as follows:

$$x_{adv} = \arg \min_{\|x'-x\|_p \leq \epsilon} \max_{\|\Delta_s\|_p \leq \tau, \, \|\Delta_g\|_p \leq \tau} d(\varphi(x_g + \Delta_g; w), \varphi(x' + \Delta_s; w)), \quad (6)$$

where $\|\cdot\|_p$ is the $\ell_p$-norm, $\epsilon$ and $\tau$ control the strength of perturbations, $\Delta_g$ and $\Delta_s$ are the data space perturbations for mitigating the approximation error in parameter space, and $x'$ denotes the perturbed version of source image. In this way, we reduce the error in parameter space to approximate the target model. We solve the inner maximization problem by generating perturbations in the feature space, given a perturbed adversarial example. This step aims to mitigate the approximation error. The outer minimization problem is solved by finding adversarial perturbations in the input space, the same as the adversarial example generation. After iterative generation of perturbations, adversarial examples are generated by attacking a model with a reduced approximation error.

## 5 Experiments

In this section, we conduct extensive experiments to verify the power of lightweight black-box attacks augmenting with ETF.

---

[2]Discussion about the assumption can be found in Appendix E.

Table 1: Performance of the different lightweight black-box and black-box adversarial attacks

| Model | VGG19 [51] | Inception v3[53] | RN152 [21] | DenseNet [23] | SENet [22] | WRN [59] | MobileNet v3[49] | Average |
|---|---|---|---|---|---|---|---|---|
| Clean | 67.43 | 64.36 | 74.21 | 73.34 | 51.28 | 73.22 | 65.06 | 66.99 |
| Autoattack[9] | 0.00 | 0.00 | 0.00 | 0.00 | 0.00 | 0.00 | 0.00 | 0.00 |
| Deep-PGD | 49.01±0.23 | 52.26±0.25 | 60.71±0.74 | 57.92±0.37 | 27.94±0.18 | 60.18±0.64 | 44.20±0.63 | 50.31±0.52 |
| Deep-MI | 38.92±0.43 | 42.37±0.37 | 49.53±0.49 | 49.06±0.89 | 19.44±0.75 | 49.11±0.82 | 33.46±0.80 | 40.69±0.96 |
| Deep-DI | 43.34±0.40 | 43.13±0.52 | 53.78±0.38 | 55.41±0.53 | 23.53±0.52 | 51.77±0.48 | 38.14±0.74 | 44.15±0.60 |
| Deep-TI | 49.46±0.52 | 49.64±0.27 | 58.89±0.71 | 58.75±0.30 | 26.19±0.16 | 56.31±0.58 | 44.02±0.46 | 49.03±0.51 |
| Shallow-PGD | 22.93±0.33 | 31.07±0.58 | 34.71±0.67 | 36.20±0.87 | 13.08±0.36 | 32.16±0.66 | 16.65±0.54 | 26.69±0.49 |
| Shallow-MI | 22.62±0.25 | 30.83±0.48 | 34.05±0.27 | 35.74±0.76 | 12.31±0.41 | 29.98±0.65 | 17.72±0.31 | 26.17±0.56 |
| Shallow-DI | 22.14±0.39 | 29.78±0.17 | 35.51±0.33 | 35.79±0.61 | 8.99±0.42 | 30.61±0.88 | 16.88±0.47 | 25.67±0.55 |
| Shallow-TI | 21.82±0.45 | 28.54±0.34 | 34.78±0.15 | 34.71±0.39 | 7.96±0.48 | 30.14±0.85 | 15.77±0.51 | 24.81±0.37 |
| ETF-PGD | **14.11**±0.24 | 20.22±0.29 | 24.20±0.34 | 24.74±0.37 | 6.96±0.44 | **20.73**±0.28 | **10.66**±0.31 | **17.37**±0.35 |
| ETF-MI | 15.32±0.52 | 19.97±0.28 | 26.25±0.14 | 28.10±0.65 | 7.02±0.43 | 22.21±0.66 | 12.23±0.32 | 18.72±0.45 |
| ETF-DI | 14.77±0.35 | 20.63±0.32 | 23.71±0.83 | 25.70±0.51 | 7.23±0.37 | 20.22±0.64 | 11.53±0.50 | 17.68±0.47 |
| ETF-TI | 15.45±0.37 | **18.03**±0.34 | **22.63**±0.45 | **24.20**±0.68 | **6.94**±0.41 | 21.53±0.25 | 12.88±0.34 | 17.38±0.71 |
| Deep*-PGD | 12.43±0.51 | 28.15±0.43 | 16.54±0.49 | 12.61±0.22 | 7.09±0.32 | 13.33±0.54 | 9.64±0.28 | 14.25±0.37 |
| Deep*-MI | 11.77±0.75 | 25.14±0.56 | 18.10±0.64 | 13.72±0.34 | 4.26±0.35 | 14.61±0.37 | 8.30±0.37 | 13.70±0.68 |
| Deep*-DI | 7.61±0.41 | 18.17±0.45 | 8.23±0.33 | 9.90±0.57 | 6.66±0.34 | 9.72±0.42 | 7.91±0.46 | 9.74±0.55 |
| Deep*-TI | 9.55±0.48 | 23.48±0.86 | 13.51±0.46 | 10.63±0.64 | 6.46±0.26 | 10.92±0.61 | 9.55±0.35 | 12.01±0.43 |

Deep* refers to the attacks mounted in the model trained on the large-scale training data.

## 5.1 Experimental Setup

**Models Architectures.** All surrogate models are based on the ResNet-18 [21]. In the ordinary settings of the black-box attack, the general surrogate models are trained through the whole training set of ImageNet [47]. On the contrary, the lightweight surrogate models adopt only $1,000$ images randomly sampled from the validation set of ImageNet, considering that images in the training set are probably inaccessible in practice [34]. Results evaluated on the CIFAR10 dataset can be found in Appendix D. Please refer to Table 5 in the Appendix B for the detailed information on the lightweight surrogate model and the general surrogate model. Regarding the target models, various model architectures are selected for full comparison, including, VGG-19 [51], Inception v3 [53], ResNet-152 [21], DenseNet [23], SENet [22], wide ResNet (WRN) [59] and MobileNet v2 [49]. All these target models are well-trained on ImageNet.

**Implement Details.** The lightweight surrogate models are randomly initialized. The batch size is $128$, the epoch is $500$, and the initial learning rate is $0.4$, linearly decreasing to $0.008$. To better approximate the low-level feature of shallow layers, we refer to the self-supervision work [5] and apply various random augmentations to the data in each epoch for training. To mount attacks, the classic methods, e.g., PGD [40], MI [11], DI [56], and TI [12], are applied to all the surrogate models. Unless otherwise specified, our attack is mounted on the first layer of ResNet-18 for lightweight surrogate models. Following previous works on intermediate-level attacks, the metric $d$ in Eq. 3 of Shallow- in Table 1 is instantiated as the $\ell_2$-norm. Meanwhile, to maximize the potential of ETFs, we apply contrastive loss to ETF-($\cdot$) in Table 1. The Top-1 prediction accuracy is adopted as the evaluation metric, i.e., lower the classification accuracy means better attack success rate. All the experiments are run for 5 individual trials with different random seeds. In our experiments, target models are evaluated using $\ell_\infty$-norm adversarial examples with maximum distortion $\epsilon = 0.1$, results evaluated under mores settings, e.g., more strict constraints $\epsilon = 0.05$, $\ell_2$-norm adversarial examples, and more test images 5,000 images, can be found in Appendix B.4.

## 5.2 Main Results

**Results on normally trained models.** The results of classification accuracy are shown in Table 1, which exhibits the evaluations for both lightweight and general surrogate models. The last four rows show the ordinary results where the general surrogate models are trained through all data in the training set. The black-box attacks are mounted based on the whole model, termed "Deep/Deep*". As the size of the training data is reduced to $1,000$, this conventional method obviously loses its aggressiveness. Specifically, the averaged accuracies of 3-6 rows are among 40-50%, revealing limited attacking capability compared to $66.99\%$ of the clean accuracy. It demonstrates the necessity

Table 2: The accuracy (%) of 7 normally trained target models evaluated on 1000 adversarial examples generated by no-box attacks and lightweight black-box attacks with different intermediate-level attacks. The best results are in **bold**. (The lower, the better)

| Model | VGG19 [51] | Inception v3[53] | RN152 [21] | DenseNet [23] | SENet [22] | WRN [59] | MobileNet v3[49] | Average |
|---|---|---|---|---|---|---|---|---|
| Clean | 67.43 | 64.36 | 74.21 | 73.34 | 51.28 | 73.22 | 65.06 | 66.99 |
| No-box[34] | 18.74 | 33.68 | 34.72 | 26.06 | 42.36 | 33.16 | 16.34 | 29.29 |
| ILA[24] | 20.13 | 28.01 | 35.72 | 35.14 | 9.16 | 29.97 | 14.31 | 24.63 |
| AA[28] | 22.76 | 31.21 | 36.67 | 37.94 | 7.72 | 33.16 | 18.17 | 26.81 |
| ETF | **14.11** | **20.22** | **24.20** | **24.74** | **6.96** | **20.73** | **10.66** | **17.37** |

of large amount of data for classic black-box attacks. In rows 7-10, shallow layers without ETF are used to generate adversarial examples. The averaged accuracies of around 25% illustrates the less dependence of shallow networks on the amount of data. In rows 11-14, attacks on shallow layers are further enhanced by ETF, decreasing the averaged accuracies to about 17%. The narrow gap between the proposed ETF and the general surrogate model indicates the potential capability of the black-box attack in the no-box settings.

**Results on Adversarially Trained Models.** Here, we consider the models that are trained through adversarial training [40], which is more persuasive to evaluate the effectiveness of attacking methods. Following [40], the target models are trained on the whole training set of ImageNet. During adversarial training, the adversarial examples are generated under $\epsilon = 0, 4/255, 8/255$, where $\epsilon$ means the $\ell_\infty$-norm constraint. Table 3 exhibits the attacking results mounted by ETF, black-box attacks (general surrogate models with all training data avaliable), and no-box attacks. For simplicity, only the results of PGD attacks are reported. As analysed above, the black-box attacks shows the best attacking results, while ETF only has a narrow gap with it. However, when $\epsilon = 4/255$ or $8/255$, ETF exceeds the black-box attacks (29.13%/26.14% vs. 48.11%/38.24%). It is interesting that black-box attacks are even worse than lightweight attacks. We believe that exploiting the phenomenon can bring something new, but we leave it as our future work due to the limited space.

## 5.3 Further Analysis and Ablation Study

**Intermediate-Level Perturbations.** Since the lightweight black-box attacks are implemented mainly on intermediate features, two classic attacks mounted on intermediate features, i.e. ILA [24] and AA [28], are selected for experiments. Comparing the ILA and AA in Table 2 and Shallow-MI/DI/TI/PGD in Table 1, the difference between their average accuracies is marginal. It indicates those intermediate-level feature pertur-

Table 3: The performance of different attacks on the adversarial trained ResNet-50 [13]. Therein, $\epsilon$ refers to the constraint $\ell_\infty$ in adversarial examples for adversarial training. The accuracy (%) is evaluated on 1000 adversarial examples. $\epsilon = 0.1$ (the lower the better). White-box refers to Auto-Attack [9].

| Adv_model | Clean | ETF ours | Black-box [40] | No-box [34] | White-box [9] |
|---|---|---|---|---|---|
| $\epsilon = 0/255$ | 69.43 | 16.97 | **8.20** | 24.53 | 0.00 |
| $\epsilon = 4/255$ | 55.62 | **29.13** | 48.11 | 39.62 | 0.00 |
| $\epsilon = 8/255$ | 41.68 | **26.14** | 38.24 | 35.87 | 0.48 |

bation has limited benefit to lightweight black-box attacks. The reason is that the intermediate-level attacks highly rely on the low-level features extracted by the lightweight model to generate adversarial examples and it can weaken the role of high-level information when attacking. The proposed ETF achieves better performance. On one hand, ETF does not rely on the features from high-level layers. On the other hand, ETF introduces a min-max scheme to mitigate the adverse impact induced by the approximation error from the shallow layers.

**Comparison with No-box Attacks.** No-box attack is the first work to explore how to mount attacks in the no-box setting. Since Li et. al [34] stated that black-box attacks do not work in this setting, surrogate models are instantiated as auto-encoders rather than models for approximating target models.

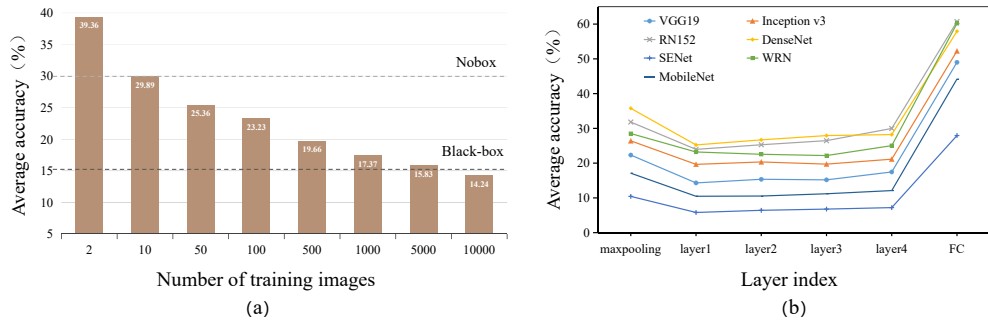

Figure 1: (a) How the lightweight attack performance of our approach varies with the number of images used for training the surrogate model. (b) The influence of low-level feature extraction at different layers of ResNet-18 on lightweight black-box attack performance. (The lower, the better)

Table 4: Model accuracy (%) under lightweight black-box attacks under challenging scenarios, where supervision information or the in-distribution data are unavailable, named Unsupervised and OOD.

| Model | VGG19 [51] | Inception v3[53] | RN152 [21] | DenseNet [23] | SENet [22] | WRN [59] | MobileNet v3[49] | Average |
|---|---|---|---|---|---|---|---|---|
| Clean | 67.43 | 64.36 | 74.21 | 73.34 | 51.28 | 73.22 | 65.06 | 66.99 |
| Supervised | 14.11 | 20.22 | **24.20** | 24.74 | 6.96 | **20.73** | 10.66 | 17.37 |
| Unsupervised | 15.54 | **19.16** | 26.27 | 23.75 | 7.66 | 22.79 | 11.43 | 18.08 |
| OOD | **6.13** | 21.72 | 25.44 | **21.89** | **5.02** | 24.33 | **7.16** | **15.96** |

Namely, they design the pretext task to train 20 auto-encoders per category to generate adversarial examples. However, the difference between the pretext tasks and the target tasks may significantly reduce the transferability of adversarial examples. In contrast, our method trains the lightweight surrogate model for tasks similar to or same as the task of target models. Hence, though only one sample per category is available, we achieve better performance than no-box attacks, as shown in Table 2.

**Number of Training Samples.** One key difference between the lightweight black-box attack and existing black-box attacks mainly lies in the number of samples $n$ used for training surrogate models. Therefore, we further explore how lightweight attack performance varies with different number of samples, i.e., $n$. We conduct experiments on randomly selected $n$ images from the validation set of ImageNet. The experimental results are shown in Figure 1. The average accuracy decreases as more samples are available, thus achieving better attacking results. More data makes it possible for the lightweight black-box attack to approximate the shallow layers of the target model more accurately. Especially the lightweight black-box attacks with $10,000$ test images can achieve the same performance as general black-box attacks which train the surrogate model on $1,200,000$ training images. Also, our method can still mount attacks even when only 2 or 10 images are available.

**Surrogate Model.** As analyzed above, we utilize the first few layers of the lightweight surrogate model to approximate the shallow layers of the target model to mount attacks. Hence, it would be interesting to study how the layers selection for lightweight models impacts the attack performance. We adopt different layers of ResNet-18 to generate adversarial examples utilizing ETF and attack the different target models. The results are summarized in Fig. 1. It can be seen that low-level information in the first block of the model is sufficient for achieving promising attack performance. With more block information on the model, the performance of the attack does not improve. In particular, the transferability of adversarial examples significantly decreases when we attack the fully-connected layer, verifying our analysis that the high-level layer of the lightweight model cannot approximate the target model well. More results for architecture selection can be found in Appendix B.4.

**Adversarial Example Visualization.** To verify that adversarial examples are truly imperceptible, we provide visualization in Figure 2, where Deep*-PGD attack (using training images), Deep-PGD attack (using test images), and lightweight black-box attack are considered.

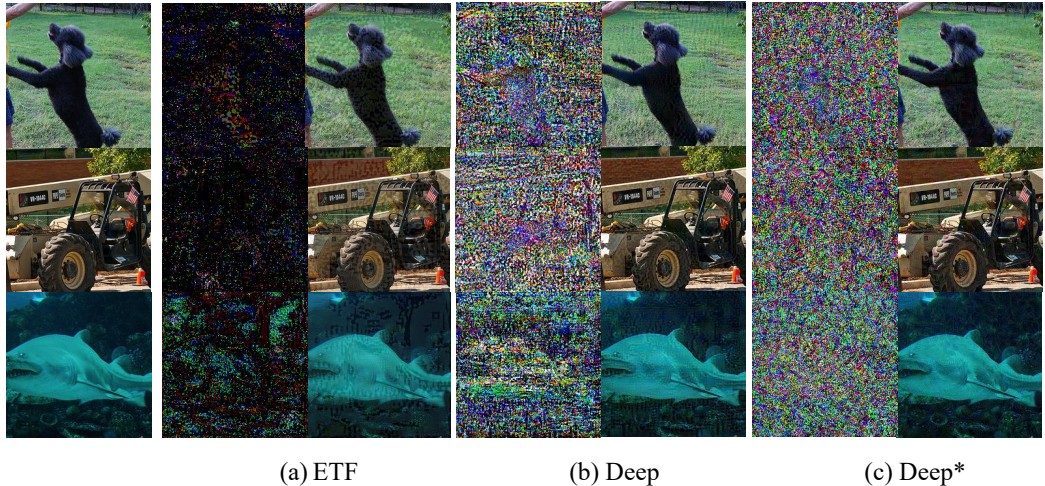

(a) ETF                   (b) Deep                   (c) Deep*

Figure 2: Adversarial examples crafted by: a) ETF, b) Deep, and c) Deep* attacks.

## 6 Discussion

Though the no-box threat model provides a weak assumption, many realistic scenarios may be more complex. For example, adversaries have access to only data without supervision information due to security [35] or privacy [46] issues. In more extreme cases, e.g., autonomous vehicle [54], adversaries even have no access to the test samples from the distribution over which the model is trained. Instead, the adversaries have access to samples from a similar (but not exact) distribution (because the adversary knows the deployment environment). Hence, we consider two more challenging scenarios in which we provide solutions to mount lightweight black-box attacks.

**Insufficient supervision information.** We design an experiment for mounting attacks using $1,000$ unlabeled samples randomly selected from the ImageNet validation set to explore the power of ETF under this challenging setting with unlabeled data. Thanks to the framework of lightweight attack, it is feasible to train the unsupervised model on unlabeled data for attacks using Eq. 2. Specifically, we adopt contrastive learning [5] to train a lightweight surrogate model on the small-scale unlabeled dataset and generate adversarial examples by attacking the trained lightweight surrogate model. The performance is shown in Table 4, where ETF still achieves similar performance without supervision information as that of the results under supervision. Intuitively, we can also employ other self-supervised learning method, e.g., rotation prediction, to train shallow layers, see details of experimental settings and results in Appendix G.

**Consideration of Out-of-distribution.** We further study the possibility for mounting attacks under a more strict scenario, where only OOD (out-of-distribution, OOD) data are available. Specifically, we utilize the model pre-trained on OOD samples to mount lightweight black-box attacks. The rationality lies in that only low-level information is required to mount lightweight black-box attacks and the acquisition of low-level information. We load pre-trained ResNet-50 on STL-10 [38] dataset as the surrogate model. Without further fine-tuning, we experiment with two test in-distribution samples, and the results are recorded in Table 4, named OOD. As we can see, the lightweight black-box attack with models trained on OOD data can achieve a surprising attack success rate, showing that black-box attacks enhanced with ETF can pose practical threats to deployed models.

## 7 Conclusion

In this paper, we propose the conception of the lightweight black-box attack to reveal the potential risk of black-box attacks under the no-box threat model. To mount effective lightweight attacks, we find it crucial to leverage DNNs' shallow layers because they can learn similar features regardless of the number of samples. This is based on the empirical observations that the model trained with limited samples can approximate the shallow layers of the target models within acceptable errors, which can be used for crafting adversarial examples. Therefore, to further enhance the performance, we propose Error TransFormer (EFT) to decrease the approximation error by transforming the approximation error in the parameter space into the feature space. The experiments show that the proposed method

achieves a surprising attack success rate under the no-box threat model using only one image per category, i.e., only $3\%$ lower than black-box attacks with complete training data of the target model.

## 8 Acknowledgements

This work was supported by NSFC No. 61872329 and No. 62222117, and the Fundamental Research Funds for the Central Universities under contract WK3490000005. YZ and BH was supported by NSFC Young Scientists Fund No. 62006202 and Guangdong Basic and Applied Basic Research Foundation No. 2022A1515011652. TL was partially supported by Australian Research Council Projects DP180103424, DE-190101473, IC-190100031, DP-220102121, and FT-220100318.

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
