## A  ETF on Hidden Layers

Before giving the details of applying ETF to hidden layers, we revisit the key identity:

$$\varphi(x; \{w^1 + w^1 A\} \cup \{w\backslash w^1\}) = g((w^1 + w^1 A)x; w\backslash w^1) = g(w^1(x + Ax); w\backslash w^1) = \varphi(x + Ax; w), \quad (7)$$

where $w^1$ stands for the first layer (convolution layer) parameters of model $\varphi$, $A$ is a transformation matrix used for perturbing $w^1$, $\{w^1 + w^1 A\} \cup \{w\backslash w^1\}$ means that the first layer's parameters are perturbed and the other layers' parameters keep the same, and $g$ is the function parameterized with $w\backslash w^1$ used for processing the first layer's outputs. Built upon Eq. 7, we have $\varphi(x; \{w^1 + w^1 A\} \cup \{w\backslash w^1\}) = \varphi(x + Ax; w)$.

The model $\varphi$ can be decomposed: $\varphi = g^l \circ \varphi^l$, where $\varphi^l(x)$ denotes the hidden feature at layer $l$. Then the output feature at layer $L$ is expressed as $\varphi^L(x; w) = g^l(w^l \varphi^l(x))$, where $w^l$ is the parameter used for processing the feature $\varphi^l(x)$. Then, we can apply the identity at layer $l$:

$$\varphi^L\left(x; \{w^l + w^l A^l\} \cup \{w\backslash w^l\}\right) = g^l\left((w^l + w^l A^l)\varphi^l(x)\right) = g^l\left(w^l(A^l \varphi^l(x) + \varphi^l(x))\right), \quad (8)$$

where $A^l$ is the transformation matrix at layer $l$. According to Eq. 8, we can transform a perturbation in the parameter space ($Aw^l$) as the perturbation in the feature space ($A\varphi^l(x)$). Thus, lightweight black-box attacks with ETF generate adversarial examples as follows:

$$x_{adv} = \arg \min_{\|x'-x\|_p \le \epsilon} \max_{\|\Delta_s^l\|_p \le \tau, \, \|\Delta_g^l\|_p \le \tau, l \in \{0,1,...,L-1\}} d(\varphi(x_g, \cup_l \Delta_g^l; w), \varphi(x', \cup_l \Delta_s^l; w)), \quad (9)$$

where $\Delta_s^l$ ($\Delta_g^l$) denotes the feature space perturbation ($l = 0$ means the data space), $\cup_l \Delta_s^l$ ($\cup_l \Delta_g^l$) stands for all perturbations in the feature space, and $\varphi(x_g, \cup_l \Delta_g^l; w)$ ($\varphi(x_g, \cup_l \Delta_g^l; w)$) stands for the output feature with perturbed features, where the feature of source (guide) image at layer $l$ is perturbed by $\Delta_s^l$ ($\Delta_g^l$). Although we can perform the error transformation in the feature space, the features obtained using weights with approximation errors make it challenging. Specifically, we merely know that the input distribution (can be seen as the feature map) is not biased, but the feature map obtained using any weights will cause bias. Thus, we merely apply ETF to the input layer and leave further exploitation of applying it to hidden layers as future work.

## B  Additional Detail Description

### B.1  Simplified Architecture

The architecture of surrogate models is modified to avoid overfitting. Considering the limited amount of data, we employ a network with a small model capacity to instantiate the feature extractor. In particular, ResNet-18 is simplified by reducing the number of blocks in each layer, i.e., only one block is used in each layer of ResNet-18.

### B.2  Classification Ability of the Surrogate Model.

To demonstrate that the lightweight black-box attack performance does not rely on the generalization of classification, we show the classification accuracy of the lightweight surrogate model used for ETF attacks in Table 5. To make the conclusion clearer, we also report the performance of a general surrogate model, which is trained on the training set of target models. We can see that the test accuracy of the lightweight surrogate model

Table 5: The classification performance of the lightweight surrogate model and the general surrogate model.

|  | Lightweight surrogate model | General surrogate model |
|---|---|---|
| Number of samples for training | 1000 | 1 200 000 |
| Training accuracy | 96.38 | 72.71 |
| Test accuracy | 2.36 | 63.24 |

(about 2) is drastically lower than that of the general model, bringing fresh air for black-box attacks. Specifically, the common sense in black-box attacks is that mounting attacks requires a surrogate model, which generalizes well on the test set. However, the experimental results in Table 5 show that the lightweight surrogate model has poor classification performance, suggesting that the generalizability of surrogate models is not a necessary condition for performing black-box attacks.

## B.3 Approximation to more layers.

Besides applying ETF in the first layer, we also conduct experiments that apply ETF to all layers except layer $L$. The results are reported in Table 6. It can be seen that applying ETF to other layers can marginally promote the attack success rate. The phenomenon may result from the fact that the approximation error can accumulate with depth. Moreover, considering that applying ETF will cause more computational overhead, we merely give the results of applying ETF to the first layer in the main paper.

Table 6: Apply ETF to all layers (except layer $L$) to further approximate the target model. This experiment is called "All" below. Similarly, "First" means only applying ETF to the first layer. (The lower, the better)

| Model | VGG19 [51] | Inception v3[53] | RN152 [21] | DenseNet [23] | SENet [22] | WRN [59] | MobileNet v3[49] | Average |
|---|---|---|---|---|---|---|---|---|
| Clean | 67.43 | 64.36 | 74.21 | 73.34 | 51.28 | 73.22 | 65.06 | 66.99 |
| No-box[34] | 18.74 | 33.68 | 34.72 | 26.06 | 42.36 | 33.16 | 16.34 | 29.29 |
| First | 14.11 | 20.22 | 24.20 | 24.74 | 6.96 | 20.73 | 10.66 | 17.37 |
| All | 13.33 | 20.21 | 22.66 | 25.49 | 5.40 | 21.74 | 9.88 | **16.95** |

## B.4 Hardware Configuration and Computation Costs.

We conduct experiments using GEFORCE RTX 2080 Ti, CPU AMD Ryzen 7 3700X @3.6 GHz. As merely 1,000 samples are required for the training of the lightweight surrogate model, the computational overhead is much less than the training of general surrogate models.

## C   Further Experiments on ImageNet

**Strict Constraint.** We conduct experiments under smaller $\epsilon$, i.e., $\epsilon = 0.05$. The results are given in Table 7, demonstrating that ETF can generate powerful adversarial examples even with meeting more strict constraints, i.e., smaller $\epsilon$.

Table 7: The accuracy of 7 normally trained target models evaluated on 1,000 adversarial examples generated by lightweight black-box attacks or existing black-box attacks, under $\epsilon \leq 0.05$. The Shallow-(PGD, MI, DI, TI) mean applying PGD, MI, DI and TI to the shallow layers of the model. Deep-(PGD, MI, DI and TI) mean applying PGD, MI, DI and TI to the model's output. EFT-(PGD, MI, DI and TI) mean applying ETF combined with PGD, MI, DI or TI to the shallow layers.

| Model | VGG19 [51] | Inception v3[53] | RN152 [21] | DenseNet [23] | SENet [22] | WRN [59] | MobileNet v3[49] | Average |
|---|---|---|---|---|---|---|---|---|
| Clean | 67.43 | 64.36 | 74.21 | 73.34 | 51.28 | 73.22 | 65.06 | 66.99 |
| Autoattack | 0.00 | 0.20 | 0.00 | 0.00 | 0.00 | 0.10 | 0.00 | 0.04 |
| Deep-PGD | 61.14 | 63.05 | 65.78 | 62.31 | 34.50 | 68.17 | 56.65 | 58.8 |
| Shallow-PGD | 46.55 | 49.13 | 56.78 | 58.34 | 28.50 | 55.82 | 37.94 | 47.58 |
| ETF-PGD | 41.76 | 46.74 | 48.55 | 50.79 | 24.68 | 53.11 | 32.65 | 42.61 |
| Deep*-PGD | 16.23 | 36.71 | 25.36 | 24.62 | 18.16 | 31.42 | 13.34 | 23.69 |

**More Validation Images.** Besides the widely used setting on the number of samples, i.e., 1,000 images, we also evaluate different methods using more samples, i.e., 5,000 images, and report the results in Table 8. The conclusion drawn from Table 8 is consistent with that drawn from Table 1, e.g., EFT outperforms "shallow" attack methods, demonstrating that ETF can generate powerful adversarial examples under various scenarios.

Table 8: The accuracy of 7 normally trained target models evaluated on 5,000 adversarial examples generated by lightweight black-box attacks or existing black-box attacks, under $\epsilon \leq 0.1$. The Shallow-(PGD, MI, DI, TI) mean applying PGD, MI, DI and TI to the shallow layers of the model. Deep-(PGD, MI, DI and TI) mean applying PGD, MI, DI and TI to the model's output. EFT-(PGD, MI, DI and TI) mean applying ETF combined with PGD, MI, DI or TI to the shallow layers. Auto-attack[23] is used for testing the robustness of the target models, so it adopts the white-box setting to mount the target models.

| Model | VGG19 [51] | Inception v3[53] | RN152 [21] | DenseNet [23] | SENet [22] | WRN [59] | MobileNet v3[49] | Average |
|---|---|---|---|---|---|---|---|---|
| Clean | 67.43 | 64.36 | 74.21 | 73.34 | 51.28 | 73.22 | 65.06 | 66.99 |
| Autoattack[9] | 0.00 | 0.00 | 0.00 | 0.00 | 0.00 | 0.00 | 0.00 | 0.00 |
| Deep-PGD | 55.86 | 56.08 | 64.48 | 65.44 | 35.92 | 63.54 | 51.10 | 56.06 |
| Deep-MI | 38.02 | 44.70 | 52.56 | 52.98 | 13.22 | 49.74 | 28.92 | 40.02 |
| Deep-DI | 51.32 | 51.10 | 61.44 | 61.60 | 33.34 | 60.36 | 47.70 | 52.41 |
| Deep-TI | 55.00 | 54.94 | 64.60 | 64.48 | 36.80 | 63.86 | 51.50 | 55.88 |
| Shallow-PGD | 19.42 | 25.12 | 31.04 | 31.70 | 9.28 | 29.16 | 16.64 | 23.19 |
| Shallow-MI | 22.47 | 28.14 | 34.69 | 35.76 | 11.42 | 31.65 | 17.13 | 25.89 |
| Shallow-DI | 19.68 | 24.62 | 30.26 | 32.17 | 10.02 | 28.24 | 16.08 | 23.01 |
| Shallow-TI | 20.40 | 23.96 | 29.00 | 31.04 | 9.82 | 28.26 | 17.08 | 22.79 |
| ETF-PGD | 13.56 | 17.66 | 23.68 | 24.60 | **4.54** | 20.68 | 9.42 | 16.31 |
| ETF-MI | 15.94 | 20.32 | 26.28 | 26.74 | 5.52 | 22.72 | 9.70 | 18.17 |
| ETF-DI | **13.16** | 25.72 | 22.32 | 22.76 | 4.68 | 19.84 | **8.58** | 15.29 |
| ETF-TI | 13.30 | **14.60** | **20.48** | **22.38** | 5.22 | **19.06** | 9.50 | **14.93** |
| Deep*-PGD | 12.43 | 28.15 | 16.54 | 12.61 | 7.09 | 13.33 | 9.64 | 14.25 |
| Deep*-MI | 11.77 | 25.14 | 18.10 | 13.72 | 4.26 | 14.61 | 8.30 | 13.70 |
| Deep*-DI | 7.61 | 18.17 | 8.23 | 9.90 | 6.66 | 9.72 | 7.91 | 9.74 |
| Deep*-TI | 9.55 | 23.48 | 13.51 | 10.63 | 6.46 | 10.92 | 9.55 | 12.01 |

$\ell_2$**-norm Perturbation.** We mainly conduct experiments with $\ell_\infty$ perturbation since it is widely adopted in many previous works [56, 25, 12]. To further demonstrate the power of our ETF, we further evaluate different methods using $\ell_2$-norm perturbation. The results are reported in Table 9, which further demonstrate the effectiveness of our proposal. Considering that $\ell_1$ and $\ell_0$ perturbations require careful design [52, 4], it is beyond the scope of this work, so we leave it as our future work.

**Architecture Selection.** We further exploit whether the architecture of surrogate models have significant impact on the performance of ETF. Specifically, we instantiate the shallow layers with different model architectures containing ResNet [21], VGG [51], and SENet [22]. The results are reported in Table 10, demonstrating that our EFT is powerful across various model architectures.

**Heavy Data Augmentation.** We follow the empirical conclusion suggested in [1], where heavy data augmentation is vital for training appropriate shallow models. Because appropriate shallow models are necessary for mounting lightweight black-box attacks, data augmentation plays a crucial role and is heavily used in our experiments. This is supported by results shown in Table 11, where we report the performance of lightweight black-box attacks with and without data augmentation.

**Capacity to Evade Adversarial Detectors.** Adversarial Detection [39, 33, 57] aims to distinguish adversarial examples from natural examples, which is also an effective way to test the robustness of adversarial attacks. Therefore, we further exploit the capacity of ETF in evading adversarial example detectors. Specifically, we employ a detection method [39, 33] to detect adversarial examples generated by different attack methods, e.g., FGSM [19], PGD [40], BIM [11], and ETF. All settings are the same as that used in the paper, and the results are reported in Table 12. We can see that ETF performs better than the baselines, i.e., having a high probability of evading detection methods.

Table 9: The classification accuracy evaluation on $\ell_2$-norm attacks. The experiment is conducted on the ImageNet validation. Following the previous work[25] about $\ell_2$-norm attacks, the maximum disturbance $\varepsilon$ is set to 16 $\sqrt[2]{N}$ where N is the dimension of input to attacks.

| Model | VGG19 [51] | Inception v3[53] | RN152 [21] | DenseNet [23] | SENet [22] | WRN [59] | MobileNet v3[49] | Average |
|---|---|---|---|---|---|---|---|---|
| Clean | 67.43 | 64.36 | 74.21 | 73.34 | 51.28 | 73.22 | 65.06 | 66.99 |
| Autoattack[9] | 0.00 | 0.00 | 0.00 | 0.00 | 0.00 | 0.00 | 0.00 | 0.00 |
| Deep-PGD | 37.73±0.31 | 42.75±0.34 | 51.04±0.77 | 51.96±0.62 | 17.48±0.34 | 50.61±0.49 | 31.07±0.55 | 40.38±0.54 |
| Deep-MI | 40.40±0.44 | 45.02±0.51 | 54.53±0.46 | 54.13±0.53 | 17.59±0.47 | 53.22±0.63 | 32.47±0.41 | 42.48±0.55 |
| Deep-DI | 38.73±0.53 | 38.63±0.49 | 50.34±0.35 | 48.79±0.48 | 17.66±0.43 | 47.53±0.57 | 27.34±0.33 | 38.43±0.47 |
| Deep-TI | 37.89±0.23 | 37.86±0.38 | 46.52±0.46 | 45.62±0.31 | 18.54±0.44 | 46.44±0.37 | 30.75±0.52 | 37.66±0.46 |
| Shallow-PGD | 25.74±0.64 | 31.51±0.56 | 44.96±0.54 | 43.72±0.55 | 8.58±0.51 | 40.62±0.24 | 18.73±0.48 | 30.55±0.55 |
| Shallow-MI | 37.46±0.94 | 42.28±0.87 | 51.56±0.79 | 50.77±0.63 | 16.58±0.67 | 52.06±0.86 | 28.02±0.74 | 39.82±0.67 |
| Shallow-DI | 28.75±0.55 | 28.36±0.64 | 38.11±0.49 | 40.23±0.41 | 15.54±0.56 | 34.42±0.75 | 24.08±0.77 | 29.93±0.66 |
| Shallow-TI | 30.28±0.36 | 31.55±0.40 | 37.69±0.39 | 38.44±0.48 | 14.52±0.48 | 35.26±0.27 | 23.54±0.19 | 30.18±0.38 |
| ETF-PGD | **22.16±0.54** | 27.03±0.36 | 34.87±0.48 | 37.94±0.59 | **11.28±0.37** | 29.63±0.41 | **16.17±0.46** | 25.58±0.28 |
| ETF-MI | 32.76±0.95 | 33.05±0.87 | 45.91±0.91 | 44.22±0.88 | 14.38±0.76 | 41.54±0.78 | 20.76±0.69 | 33.23±0.74 |
| ETF-DI | 23.71±0.46 | **23.45±0.55** | **33.29±0.56** | **34.25±0.49** | 12.49±0.34 | **29.23±0.24** | 18.54±0.48 | **24.99±0.53** |
| ETF-TI | 25.23±0.37 | 25.73±0.68 | 34.15±0.73 | 37.34±0.43 | 12.56±0.66 | 30.07±0.56 | 21.53±0.45 | 26.65±0.69 |
| Deep*-PGD | 7.65±0.42 | 22.88±0.34 | 11.44±0.12 | 11.23±0.44 | 4.56±0.71 | 9.69±0.78 | 8.03±0.46 | 10.78±0.45 |
| Deep*-MI | 11.26±0.65 | 26.08±0.92 | 17.47±0.34 | 15.73±0.56 | 4.78±0.48 | 14.52±0.41 | 8.58±0.88 | 14.06±0.57 |
| Deep*-DI | 1.04±0.34 | 11.04±0.54 | 1.68±0.48 | 1.39±0.51 | 0.77±0.32 | 3.01±0.29 | 0.56±0.41 | 2.78±0.42 |
| Deep*-TI | 5.56±0.44 | 18.09±0.36 | 9.94±0.43 | 10.42±0.37 | 3.23±0.74 | 8.27±0.43 | 6.54±0.43 | 8.86±0.49 |

Table 10: Model accuracy under ETF attack with different architectures, containing SENet, VGG11, and ResNet18.

| Model | VGG19 [51] | Inception v3[53] | RN152 [21] | DenseNet [23] | SENet [22] | WRN [59] | MobileNet v3[49] | Average |
|---|---|---|---|---|---|---|---|---|
| Clean | 67.43 | 64.36 | 74.21 | 73.34 | 51.28 | 73.22 | 65.06 | 66.99 |
| SENet [22] | 23.44 | 28.42 | 35.07 | 31.64 | 6.73 | 28.19 | 11.80 | 23.61 |
| VGG11 [51] | 18.20 | 22.65 | 27.24 | 26.33 | **6.47** | 23.16 | 12.69 | 19.53 |
| Resnet [21] | **14.11** | **20.22** | **24.20** | **24.74** | 6.96 | **20.73** | **10.66** | **17.37** |

Table 11: The impact of augmentation to ETF attacks. "No-Aug" means the effect of the attack on the ETF using the surrogate model without augmentation for training. This experiment is conducted on the ImageNet validation. The best results are in bold.

| Model | VGG19 [51] | Inception v3[53] | RN152 [21] | DenseNet [23] | SENet [22] | WRN [59] | MobileNet v3[49] | Average |
|---|---|---|---|---|---|---|---|---|
| Clean | 67.43 | 64.36 | 74.21 | 73.34 | 51.28 | 73.22 | 65.06 | 66.99 |
| No-Aug | 34.58 | 39.17 | 46.25 | 50.06 | 10.42 | 45.10 | 22.92 | 35.50 |
| Aug | **14.11** | **20.22** | **24.20** | **24.74** | **6.96** | **20.73** | **10.66** | **17.37** |

Table 12: Performance of adversarial detection against four attacks, metric to evaluate the detection performance can be found in [33, 39].

| Mahalanobis [33] | | | | | |
|---|---|---|---|---|---|
| Method | TNR | AUROC | DTACC | AUIN | AUOUT |
| BIM [32] | 99.99 | 99.99 | 99.86 | 99.86 | 99.71 |
| FGSM [19] | 98.89 | 99.88 | 98.89 | 99.66 | 99.24 |
| Deep*-PGD [40] | 97.22 | 99.58 | 97.92 | 99.64 | 99.05 |
| ETF | **96.67** | **98.73** | **96.94** | **98.75** | **97.98** |
| LID[39] | | | | | |
| Method | TNR | AUROC | DTACC | AUIN | AUOUT |
| Deep*-BIM [32] | 99.99 | **98.81** | 98.33 | 99.77 | 99.33 |
| Deep*-FGSM [19] | 99.99 | 99.99 | 99.99 | 99.72 | 99.44 |
| Deep*-PGD [40] | 99.99 | 99.99 | 99.99 | 99.86 | 99.72 |
| ETF | **97.78** | 99.58 | **97.22** | **99.51** | **98.68** |

# D  Results on CIFAR10

We conduct the experiments on the CIFAR10 dataset, see Table 13, and evaluate the robustness of models downloaded from RobustBench [8], see Table 14. The conclusion drawn from Table 13 and Table 14 is consistent with that drawn from Table 1 evaluating on ImageNet dataset.

Table 13:  Evaluate the performances of different attacks on CIFAR10. Here, experiments of "Deep-, Shallow-, ETF-" are conducted in the no-box threat model. "Deep*" means the black-box setting where the surrogate models are trained on the training data the same as the seven target models. "PGD [40], MI [11], DI [56], TI [12]" is applied to the different settings and methods. Auto-attack[23] is used for testing the robustness of the target models, so it adopts the white-box setting to mount the seven target model. $\varepsilon \leq 0.1$ in $\ell_\infty$-norm.

| Model | VGG19[51] | RN56[21] | MobileNet[49] | ShuffleNet[22] | Avg |
|---|---|---|---|---|---|
| clean | 93.91 | 94.37 | 93.72 | 92.98 | 93.74 |
| Auto-attack [9] | 0.00 | 0.00 | 0.00 | 0.00 | 0.00 |
| Deep-PGD | 59.45 ±0.34 | 57.58 ±0.46 | 45.21 ±0.27 | 52.32 ±0.37 | 53.64 ±0.78 |
| Deep-MI | 53.44 ±0.75 | 52.17 ±0.65 | 44.25 ±0.34 | 49.80 ±0.35 | 49.91 ±0.58 |
| Deep-DI | 60.24 ±0.19 | 58.63 ±0.34 | 47.67 ±0.31 | 54.34 ±0.62 | 55.22 ±0.52 |
| Deep-TI | 64.51 ±0.38 | 59.85 ±0.60 | 48.80 ±0.59 | 56.88 ±0.44 | 57.51 ±0.42 |
| Shallow-PGD | 27.17 ±0.74 | 31.06 ±0.55 | 22.83 ±0.66 | 28.14 ±0.76 | 27.30 ±0.81 |
| Shallow-MI | 32.43 ±0.98 | 36.42 ±1.01 | 31.84 ±0.79 | 30.76 ±0.94 | 32.86 ±0.94 |
| Shallow-DI | 25.65 ±0.56 | 30.27 ±0.51 | 22.61 ±0.38 | 27.22 ±0.55 | 26.43 ±0.45 |
| Shallow-TI | 28.66 ±0.45 | 31.35 ±0.33 | 27.20 ±0.44 | 29.48 ±0.63 | 29.17 ±0.56 |
| ETF-PGD | 21.27 ±0.27 | 25.85 ±0.84 | **20.03** ±0.65 | 22.37 ±0.44 | 22.38 ±0.53 |
| ETF-MI | **20.75** ±0.55 | **24.36** ±0.35 | 20.51 ±0.34 | **19.68** ±0.23 | **21.32** ±0.42 |
| ETF-DI | 21.37 ±0.37 | 26.46 ±0.27 | 21.11 ±0.69 | 23.14 ±0.36 | 23.02 ±0.55 |
| ETF-TI | 25.48 ±0.41 | 30.26 ±0.23 | 23.37 ±0.51 | 26.34 ±0.25 | 26.36 ±0.39 |
| Deep*-PGD | 4.63 ±0.54 | 0.81 ±0.74 | 3.79 ±0.28 | 3.21 ±0.32 | 3.11 ±0.47 |
| Deep*-MI | 4.72 ±0.20 | 0.96 ±0.36 | 4.36 ±0.12 | 3.78 ±0.25 | 3.45 ±0.33 |
| Deep*-DI | 4.63 ±0.17 | 0.81 ±0.67 | 2.38 ±0.53 | 3.34 ±0.43 | 2.79 ±0.47 |
| Deep*-TI | 4.66 ±0.18 | 0.84 ±0.25 | 3.78 ±0.46 | 3.67 ±0.31 | 3.23 ±0.32 |

Table 14: The attacks on the most robust models from CIFAR10 RobustBench. The robustness model is trained by the different adversarial defense method,$\varepsilon \leq 0.1$ in $\ell_\infty$-norm.

| Model | Gowal2021 [20] | Kang2021 [29] | Pang2022 [42] | Sehwag2021 [50] | Avg |
|---|---|---|---|---|---|
| clean | 89.00 | 92.00 | 87.50 | 86.50 | 88.75 |
| Auto-attack [9] | 0.00 | 0.00 | 0.00 | 0.00 | 0.00 |
| ETF-PGD | 72.01 | 72.86 | 72.50 | 67.44 | 71.20 |
| Deep*-PGD | 83.53 | 88.06 | 83.17 | 79.44 | 83.55 |

## E  Target Model Approximation Assumption

Taking the first layer as an example, let $w_t^1$ and $w^1$ stand for the parameters of the target and surrogate models, respectively. In many practical scenarios, $w_t^1$ and $w^1$ usually have different dimensions, leading to intractable parameters' discrepancy alleviation. Fortunately, we can find an appropriate low-rank approximation for parameters of deep neural networks [10, 62, 30]. Specifically, we can approximate either $w_t^1$ or $w^1$ to make these two matrices have the same dimensions, so we can consider that the dimensions of the two models are the same. Consequently, we can find a transformation matrix $A$ such that the approximation error is minimized, i.e., $A = \arg\min_{\tilde{A}} |w_t^1 - w^1 - w^1\tilde{A}|_F$ , where $|\cdot|_F$ is the Frobenius norm. In this paper, we assume the approximation error is infinitesimal, i.e., $|w_t^1 - w^1 - w^1 A|_F = 0$. Then, we leverage $w^1$ and $A$ to represent the target model, i.e., $w_t^1 = w^1 + w^1 A$.

## F  Perturbation in Different Space

In the no-box setting, performing the min-max strategy in the feature space is more appropriate than the weight space optimization [55] for the no-box threat model. This is because we know which perturbations are preferred in the feature space, e.g., towards features of guide images, but we have no idea about which perturbations are preferred in the weight space, i.e., no "guide models", which is supported by our experiments, see Table 15.

Table 15: The model accuracy under ETF attacks, where "Feature_space" means the feature space perturbation and "Weight_space" represents the min-max strategy in the weight space [55]. This experiment is conducted on the ImageNet validation. The best results are in bold.

| Model | VGG19 [51] | Inception v3[53] | RN152 [21] | DenseNet [23] | SENet [22] | WRN [59] | MobileNet v3[49] | Average |
|---|---|---|---|---|---|---|---|---|
| Clean | 67.43 | 64.36 | 74.21 | 73.34 | 51.28 | 73.22 | 65.06 | 66.99 |
| Weight-space | 29.43 | 32.44 | 40.11 | 41.88 | 10.12 | 35.41 | 19.27 | 29.81 |
| Feature-space | **14.11** | **20.22** | **24.20** | **24.74** | **6.96** | **20.73** | **10.66** | **17.37** |

## G  Different Self-supervised Learning Approach

It is straightforward that exploring different strategies to train the shallow model is exciting for further improvement of the performance of lightweight black-box attacks, as shallow layers play an important role in lightweight black-box attacks. Thus, we generate adversarial examples using EFT with shallow layers trained with a rotation prediction task [17] and report the results in Table 16. We can see that shallow layers trained with the rotation prediction task is slightly worse than using the contrastive strategy, but the performance can also reduce the model accuracy significantly.

## H  Social Impact

The motivation of this work is to provide an approach to evaluate the adversarial robustness in a more practical scenario, the no-box setting. Defenses can be assessed with fewer constraints

Table 16: The model accuracy under ETF attacks, where "rotation" means that the shallow layers are trained using the rotation task and "classification" represents that shallow layers are trained with the classification task. This experiment is conducted on the ImageNet validation. The best results are in bold.

| Model | VGG19 [51] | Inception v3[53] | RN152 [21] | DenseNet [23] | SENet [22] | WRN [59] | MobileNet v3[49] | Average |
|---|---|---|---|---|---|---|---|---|
| Clean | 67.43 | 64.36 | 74.21 | 73.34 | 51.28 | 73.22 | 65.06 | 66.99 |
| Rotation [17] | 19.07 | 21.79 | 27.30 | 28.85 | 7.66 | 23.94 | 12.51 | 20.16 |
| Classification | **14.11** | **20.22** | **24.20** | **24.74** | **6.96** | **20.73** | **10.66** | **17.37** |

through lightweight black-box attacks, i.e., without accessing training samples and any queries. We can develop defensive models robust against lightweight black-box attacks and attack algorithms to mislead deployed models. We believe the development of lightweight black-box attacks can help better access the robustness of deployed models and hope the proposed ETF can promote the development of corresponding defense methods.