# OpenReview forum: "Towards Lightweight Black-Box Attack Against Deep Neural Networks"
_NeurIPS.cc/2022/Conference — NeurIPS 2022 Accept_

### Official Review · Reviewer_kZtY · 2022-06-30

**Rating:** 7
**Confidence:** 3
**Ethics Flag:** Yes
**Soundness:** 3 good
**Presentation:** 3 good
**Contribution:** 3 good

**Summary:**

This paper studies lightweight black-box attacks with limited knowledge of the target model and its output.
The author proposed an Error Transformer(ETF) to alleviate the approximation error between the surrogate model and the target model.
The experiment results show that using ETF + limited samples is only 3% attack success rate lower than the black-box attack with full access of the training data.

**Questions:**

1.  According to eq. 4, is the Error Transformer simply learning one more transformation matrix $A$? If it is, then is $\Delta_s = Ax$? Or does it shows that the feature space perturbation at the first layer is equivalent to data space perturbation(when using $Ax$ as noise)?
2. In Section 4.3, line 185, the author said, "Therefore, connecting the parameter space ..." is there a more intuitive explanation for it? It is clear that eq. 4 shows it, but I don't think it is intuitive with the description in lines 183~184.
3. In section 4.3, lines 199~200, the author assumes there exists a transformation matrix $A$, $s.t.$ $w_t^1 = w^1 + w^1A$. How strong is the assumption here? What if the target and surrogate models have different dimensions at the first layer?
4. In the eq. 6 and 9, what does $\Delta_s$ means? I am unsure if there is a difference between $\Delta_t$ and $\Delta_s$.
5. Where is the $x^{\prime}$ defined in the paper? It looks like the perturbed version of $x$ in eq. 3, and it looks like source $x$ in eq. 6.

**Ethics Review Area:**

["Privacy and Security (e.g., consent)"]

**Limitations:**

The author addressed the social impact in the appendix.

**Strengths And Weaknesses:**

Strengths:

 1. The experiment results are compelling; this paper shows some interesting results that a black-box attack can be performed without using the training set.
 2. To my best understanding, using the shallow layer & ETF for approximation of the target model is novel to the ML security field.
 3. The motivation of the proposed method and the lightweight black-box attack are well illustrated.
 4. The author conducted a thorough analysis and ablation study in section 5.

Weaknesses:
  1.  I suggest the authors can show at least one visual example/noise generated by their method.
  2. Please refer to the questions.

I suggest the authors add more explanations to their equations to make them easier to follow. Overall, I think the experiment results are good and the motivation is clear. Therefore, I tend to vote for acceptance of this paper and am willing to change my score after the author clarifies my questions.

---

> ### Author Response · Authors · 2022-08-02
> **Response to Reviewer kZtY  (Part 2/2)**
>
>
> > Q3: "the author assumes there exists a transformation matrix. How strong is the assumption here? What if the target and surrogate models have different dimensions at the first layer?"
>
> **A3**:
> Your valuable questions make us aware that the assumption here lacks the necessary explanation, so we have added the following into the revision.
>
> Taking the first layer as an example, let $w^1_t$ and $w^1$ stand for the parameters of the target and surrogate models, respectively. In many practical scenarios, $w^1_t$ and $w^1$ usually have different dimensions, leading to intractable parameters’ discrepancy alleviation. Fortunately, we can find an appropriate low-rank approximation for parameters of deep neural networks [1,2,3]. Specifically, we can approximate either $w^1_t$ or $w^1$ to make these two matrices have the same dimensions, so we can consider that the dimensions of the two models are the same. Consequently, we can find a transformation matrix $A$ such that the approximation error is minimized, i.e.,
> $A =   \arg\min\_{\tilde{A}}|w^1\_t -w^1-w^1\tilde{A}|\_{F}$ ,    where $|\cdot|_{F}$ is the Frobenius norm.  In this paper, we assume the approximation error is infinitesimal, i.e., $|w^1\_t - w^1 - w^1A|\_{F} = 0$. Then, we leverage $w^1$ and $A$ to represent the target model, i.e., $w^1\_t = w^1 + w^1 A$.
>
> > Q4: "In the eq. 6 and 9, what does $\Delta_s$ means? I am unsure if there is a difference between $\Delta_s$ and $\Delta_t$."
>
> **A4**:
> Sorry for the typo. We have fixed it in the revision. $\Delta_s$ and $\Delta_g$ stand for the data space perturbation applied to the source and guide images, respectively. These perturbations are designed to mitigate the approximation error in parameter space.
>
> > Q5: "Where is the $x'$ defined in the paper? It looks like the perturbed version of $x$ in eq. 3, and it looks like source $x$ in eq. 6."
>
> **A5**:
> Thank you for pointing out the confusing problem. We have fixed it in the revision. $x'$ denotes the perturbed version of source image $x$ in Eq. (3) and Eq. (6). In Eq. (6), $x' + \Delta_s$ stands for the perturbing operation applied to the perturbed input $x'$, where $x'$ is obtained by perturbing $x$ for optimizing the adversarial loss while $x'$ is perturbed by $\Delta_s$ to mitigate the approximation error.
>
> **Reference**
>
> [1] Exploiting Linear Structure Within Convolutional Networks for Efficient Evaluation. NeurIPS 2014.
>
> [2] Accelerating Very Deep Convolutional Networks for Classification and Detection. TPAMI 2015.
>
> [3] Compression of Deep Convolutional Neural Networks for Fast and Low Power Mobile Applications. ICLR 2016.

---

> > ### Comment · Reviewer_kZtY · 2022-08-08
> > **Re: Response to Reviewer kZtY**
> >
> > Thank you for the response. I think my questions and concerns were addressed and solved.
> > However, I can't find the revision version of the paper, have the authors uploaded the revision yet?

---

> > > ### Author Response · Authors · 2022-08-08
> > > **Glad to hear that your concerns and questions are addressed well!**
> > >
> > > Dear Reviewer kZtY,
> > >
> > > We are so glad that our response clarified your questions and concerns. Thanks for your kind reminder for uploading the revision, where we carefully revised the paper according to reviewers' constructive suggestions and comments. The revised version has been uploaded and the modified content was marked in blue.
> > >
> > > Best regards,
> > >
> > > Authors of #565

---

> > > ### Author Response · Authors · 2022-08-09
> > > **Would you mind checking whether you received the revision?**
> > >
> > > Dear Reviewer kZtY,
> > >
> > > Would you mind re-checking the system to confirm whether you have received the latest version of our paper?
> > >
> > > We revised the paper following your constructive suggestions, making the paper more readable and solid. Thus, we look forward to your further outstanding questions/comments.
> > >
> > > If you find that the latest revision meets the bar to change your score, would you mind raising the score? Your support for a novel, simple, and initial attempt to think of the potential threats of black-box attacks is critical, and we sincerely appreciate it!
> > >
> > > Best regards,
> > >
> > > Authors of #565

---

> ### Author Response · Authors · 2022-08-02
> **Response to Reviewer kZtY (Part 1/2)**
>
>
> **Response to Reviewer kZtY**
>
> We sincerely thank you for taking the time to review our paper carefully and for your constructive comments and positive feedback about our work. Please find our responses below.
>
> **Response to [Weakness]**
>
> > W1: "I suggest the authors can show at least one visual example/noise generated by their method."
>
> **R1:**
> Thanks for your valuable suggestion. We have added a figure to the revision, where adversarial examples and perturbations are generated by different methods, containing deep*-PGD attack using training images, deep-PGD attack using test images, and lightweight black-box attack.
>
> > W2: "I suggest the authors add more explanations to their equations to make them easier to follow. "
>
> **R2:**
> Following your suggestion, we have provided more details to explain and describe all equations in the revision.
>
>
> **Response to [Questions]**
>
> > Q1: "According to eq. 4, is the Error Transformer simply learning one more transformation matrix $A$? If it is, then is
> $\Delta_s=Ax$? Or does it shows that the feature space perturbation at the first layer is equivalent to data space perturbation (when using $Ax$ as noise)?"
>
> **A1:**
> Your understanding is correct. The implicitly learned matrix $A$ connects the feature space perturbation and weight space perturbation, so that we can transform the approximation error ($wA$) in the weight space to the feature (or input) space, i.e., $\Delta_s=Ax$. Thus, the data space perturbation is equivalent to perturbing the first layer parameters, i.e., correcting the first layer parameters by adding perturbations in the data space.
>
> > Q2: "In Section 4.3, line 185, the author said, "Therefore, connecting the parameter space ..." is there a more intuitive explanation for it? It is clear that eq. 4 shows it, but I don't think it is intuitive with the description in lines 183-184."
>
> **A2:**
> Thanks for your insightful question! The question makes us aware that exploring which kind of feature perturbations are more preferred is exciting and interesting, which can benefit the attack success rate of lightweight black-box attacks. Therefore, we have added the following description to our revision.
>
> To alleviate the approximation error of shallow models, we propose transforming the parameter space's approximation error as the feature space's perturbation. The inspiration is borrowed from the feature space attack. Specifically, we have little knowledge to determine which perturbations can point (from the surrogate model) to the target model, making it challenging to alleviate the approximation error in the weight space. In contrast, we have the prior that samples with different labels should have distinguishable representations/features. Thus, we can leverage the prior knowledge to select preferred perturbations in the feature space, i.e., we prefer perturbations that can make representations/features of samples with different labels indistinguishable. Therefore, we design a min-max optimization to identify the "worst'' model, and then make different image features obtained by the worst model indistinguishable. Consequently, we select a guide image for each source image and generate adversarial examples by perturbing the source image to make the guide and source images have the same/similar representation/features.
>
> Inspired by the question, we are aware that how selecting a guide image is an exciting direction to improve the performance of lightweight black-box attacks further.

---

### Official Review · Reviewer_RUwr · 2022-07-09

**Rating:** 5
**Confidence:** 3
**Soundness:** 3 good
**Presentation:** 2 fair
**Contribution:** 3 good

**Summary:**

The authors propose a "lightweight" black-box attack method that uses limited data called Error TransFormer (ETF). Experimental results show that the proposed method achieves decent performance compared to other black-box attacks with few available examples.

**Questions:**

I have no questions for now.

**Limitations:**

I didn't find any potential negative societal impact.

**Strengths And Weaknesses:**

Strengths:
+ The method is novel to me, and technically sound.
+ The performance is good. It is interesting that only 2 training images can still drop large accuracy.

Weakness:
- The writing should be carefully checked. I notice there are many errors that may lead the reader to significant confusion, for instance, the distance should be *maximize* instead of *minimize* for generating adversarial examples in line 173, the similarity should be *decrease* in line 234, the first row in Table 1 has some error, and it should be $\epsilon=0.1$ in the caption of Table 1. Hope the authors check the paper carefully and fix these errors.
- The authors only test under $\epsilon = 0.1$, it is more convincing that perform some experiments under smaller $\epsilon$, such as $\epsilon=0.05$.
- It is more convincing that perform experiments on more examples since the size of only 1,000 examples is too small.
- It is better bold the best results in *all tables*.
- The experiment setup should be more detailed.

---

> ### Author Response · Authors · 2022-08-02
> **Response to Reviewer RUwr (Part 2/2)**
>
> > W3: "It is more convincing that perform experiments on more examples since the size of only 1,000 examples is too small."
>
> **R3:**
> Following your kind suggestion, we evaluate different methods using more samples, i.e., 5000 images, and report the results in [TABLE 3-2]. The conclusion drawn from [TABLE 3-2] is consistent with that drawn from Table 1 (in the paper), e.g., EFT outperforms "shallow'' attack methods, demonstrating that ETF can generate powerful adversarial examples under various scenarios.
>
>
> **TABLE 3-2:** The accuracy of 7 normally trained target models evaluated on 5,000 adversarial examples generated by lightweight black-box attacks or existing black-box attacks, under $\epsilon \leq 0.1$. The Shallow-(PGD, MI, DI, TI) mean applying PGD, MI, DI and TI to the shallow layers of the model. Deep-(PGD, MI, DI and TI) mean applying PGD, MI, DI and TI to the model’s output. EFT-(PGD, MI, DI and TI) mean applying ETF combined with PGD, MI, DI or TI to the shallow layers. Auto-attack[23] is used for testing the robustness of the target models, so it adopts the white-box setting to mount the  target models.
>
> | Model         | VGG19[1]      | Inception_v3[2] | RN152[3]      | DenseNet161[4] | SENet[5]     | WRN[6]        | MobileNet[7]  | Avg        |
> |---------------|------------|--------------|------------|-------------|-----------|------------|-----------|------------|
> | clean         | 67.43%     | 64.35 %      | 74.21 %    | 73.34%      | 51.28%    | 73.22%     | 65.06%     | 66.99%     |
> | Deep-PGD      | 55.86%     | 56.08%       | 64.48%     | 65.44%      | 35.92%    | 63.54%     | 51.10%    | 56.06%     |
> | Deep-MI       | 38.02%     | 44.70%       | 52.56%     | 52.98%      | 13.22%    | 49.74%     | 28.92%    | 40.02%     |
> | Deep-DI       | 51.32%     | 51.10%       | 61.44%     | 61.60%      | 33.34%    | 60.36%     | 47.70%    | 52.41%     |
> | Deep-TI       | 55.00%     | 54.94%       | 64.60%     | 64.48%      | 36.80%    | 63.86%     | 51.50%    | 55.88%     |
> |  Shallow-PGD  | 19.42%     | 25.12%       | 31.04%     | 31.70%      | 9.28%     | 29.16%     | 16.64%    | 23.19%     |
> | Shallow-MI    | 22.47%     | 28.14%       | 34.69%     | 35.76%      | 11.42%    | 31.65%     | 17.13%    | 25.89%     |
> | Shallow-DI    | 19.68%     | 24.62%       | 30.26%     | 32.17%      | 10.02%    | 28.24%     | 16.08%    | 23.01%     |
> | Shallow-TI    | 20.40%     | 23.96%       | 29.00%     | 31.04%      | 9.82%     | 28.26%     | 17.08%    | 22.79%     |
> | ETF-PGD       | 13.56%     | 17.66%       | 23.68%     | 24.60%      | **4.54%** | 20.68%     | 9.42%     | 16.31%     |
> | ETF-MI        | 15.94%     | 20.32%       | 26.28%     | 26.74%      | 5.52%     | 22.72%     | 9.70%     | 18.17%     |
> | ETF-DI        | 13.16%     | 25.72%       | 22.32%     | 22.76%      | 4.68%     | 19.84%     | **8.58%** | 15.29%     |
> | ETF-TI        | **13.30%** | **14.60%**   | **20.48%** | **22.38%**  | 5.22%     | **19.06%** | 9.50%     | **14.93%** |
> | Deep*-PGD     | 12.43%     | 28.15%       | 16.54%     | 12.61%      | 7.09%     | 13.33%     | 9.64%     | 14.25%     |
> | Deep*-MI      | 11.77%     | 25.14%       | 18.10%     | 13.72%      | 4.26%     | 14.61%     | 8.30%     | 13.70%     |
> | Deep*-DI      | 7.61%      | 18.17%       | 8.23%      | 9.90%       | 6.66%     | 9.72%      | 7.91%     | 9.74%      |
> | Deep*-TI      | 9.55%      | 23.48%       | 13.51%     | 10.63%      | 6.46%     | 10.92%     | 9.55%     | 12.01%     |
> | Autoattack[8]   | 0.00%      | 0.00%        | 0.00%      | 0.00%       | 0.00%      | 0.00%      | 0.00%      | 0.00%       |
>
>
> > W4: "It is better bold the best results in all tables."
>
> **R4:**
> Following your suggestion, we have bolded the best results in all tables in the revision.
>
> > W5: "The experiment setup should be more detailed."
>
> **R5:**
> Sorry for the confusion. Following your kind suggestion, we provide detailed explanations and descriptions of the experimental settings in the revision.
>
> **Reference**
>
> [1] Very deep convolutional networks for large-scale image recognition. Simonyan et al. ICLR 2015.
>
> [2] Rethinking the inception architecture for computer vision.  Szegedy  et al. CVPR 2016.
>
> [3] Deep residual learning for image recognition. He et al. CVPR 2016.
>
> [4] Densely connected convolutional networks. Huang et al. CVPR 2017.
>
> [5] Squeeze-and-excitation networks. Hu  et al.  CVPR 2018.
>
> [6] Wide residual networks. Zagoruyko  et al.   BMVC  2016.
>
> [7] Mobilenetv2: Inverted residuals and linear bottlenecks. Sandler et al.  CVPR 2018.
>
> [8]  Reliable evaluation of adversarial robustness with an ensemble of diverse parameter-free attacks.  Croce  et al. ICML 2022.

---

> ### Author Response · Authors · 2022-08-02
> **Response to Reviewer RUwr (Part 1/2)**
>
>
> **Response to Reviewer RUwr**
>
> We sincerely thank you for taking the time to review our paper carefully and for your constructive comments and positive feedback about our work. Please find our responses below.
>
> **Response to [Weakness]**
>
> > W1: "Errors about distance and captions. "Hope the authors check the paper carefully and fix these errors."
>
> **R1:**
> Sorry for our unclear description, we will definitely improve our writing and paper organization in our revision.
>
> > W2: "it is more convincing that perform some experiments under smaller $\epsilon$."
>
> **R2:**
> Following your kind suggestion, we conduct experiments under smaller $\epsilon$, i.e., $\epsilon=0.05$. The results are given in [TABLE 3-1], demonstrating that ETF can generate powerful adversarial examples even with meeting more strict constraints, i.e., smaller $\epsilon$. The results and corresponding analysis have been added to the revision.
>
>
> **TABLE 3-1:** The accuracy of 7 normally trained target models evaluated on 1,000 adversarial examples generated by lightweight black-box attacks or existing black-box attacks, under $\epsilon \leq 0.05$. The Shallow-(PGD, MI, DI, TI) mean applying PGD, MI, DI and TI to the shallow layers of the model. Deep-(PGD, MI, DI and TI) mean applying PGD, MI, DI and TI to the model’s output. EFT-(PGD, MI, DI and TI) mean applying ETF combined with PGD, MI, DI or TI to the shallow layers.
>
> | Model         | VGG19[1]      | Inception_v3[2] | RN152[3]      | DenseNet161[4] | SENet[5]     | WRN[6]        | MobileNet[7]  | Avg        |
> |--------------|------------|--------------|------------|-------------|------------|------------|------------|-------------|
> | clean         | 67.43%     | 64.35 %      | 74.21 %    | 73.34%      | 51.28%    | 73.22%     | 65.06%     | 66.99%     |
> | Deep-PGD     | 61.14%     | 63.05%       | 65.78%     | 62.31%      | 34.50%     | 68.17%     | 56.65%     | 58.8%       |
> | Shallow-PGD  | 46.55%     | 49.13%       | 56.78%     | 58.34%      | 28.50%     | 55.82%     | 37.94%     | 47.58%      |
> | ETF-PGD      | **41.76%** | **46.74%**   | **48.55%** | **50.79%**  | **24.68%** | **53.11%** | **32.65%** | **42.61%** |
> | Deep*-PGD    | 16.23%     | 36.71%       | 25.36%     | 24.62%      | 18.16%     | 31.42%     | 13.34%     | 23.69%      |
> | Autoattack[8]   | 0.00%      | 0.00%        | 0.00%      | 0.00%       | 0.00%      | 0.00%      | 0.00%      | 0.00%       |

---

> ### Author Response · Authors · 2022-08-08
> **Looking forward to your responses or further suggestions/comments!**
>
> Dear reviewer RUwr,
>
> Thank you again for your time in reviewing our paper and your constructive comments on our work. We’d be grateful if you can confirm whether our response has addressed your concerns. Here is a short summary:
>
> Inspired by your constructive questions and suggestions, we have added two tables in the revision for verifying the effectiveness of ETF under a) smaller $\epsilon$ and b) more test images. In addition, we revised our paper to improve the clarity and readability following your valuable suggestions.
>
>
> We’d be glad to answer any outstanding questions and look forward to any further discussions.
>
> Best regards,
>
> Authors of #565

---

### Official Review · Reviewer_SEqv · 2022-07-10

**Rating:** 6
**Confidence:** 4
**Soundness:** 2 fair
**Presentation:** 1 poor
**Contribution:** 3 good

**Summary:**

Recently, there has been a lot of focus on techniques for the black-box generation of adversarial attacks. However, in practice, an adversary might not have unrestricted access to the predictive interface of the target model or unlimited data to generate adversarial examples. In this paper, the authors propose Error TransFormer (ETF) for generating adversarial examples, which works even when the adversary doesn't have access to training data and model outputs. ETF works by constructing a surrogate model with the limited available data and then perturbing the features obtained from its shallow layers.

**Questions:**

- How imperceptible are the examples generated by the proposed technique? For instance, even though most black-box adversarial techniques have higher success rates, they generally forgo imperceptibility.
- How good are examples generated by the proposed technique in evading the recent class of adversarial example detection methods?
- Does surrogate model architecture impact the success rate of the proposed technique?
- Does the technique also work in other domains like NLP?
- The surrogate model is trained in a contrastive manner. Can other self-supervision tasks like rotation be used to train it? (ref. Unsupervised Representation Learning by Predicting Image Rotations)
- For table 1, were the surrogate model trained using labels or in a contrastive manner?
- How useful are the examples generated by ETF in improving the robustness of the models?
- In addition, as the performance of the surrogate is tied to the samples used for training it, I would encourage the authors to run the experiments on different sample sets and report the variance.

**Limitations:**

I would encourage the authors to list the limitations of the proposed approach.

**Strengths And Weaknesses:**

Strengths:
- There are relatively few techniques that can work under the threat model considered in the paper.
- Even though the technique appears simple, it performs pretty well in practice.


Weaknesses:
- The paper is a bit hard to follow, and the writing needs improvement.
- I found the experiments a bit lacking, and please refer to the next section for a list of questions I would like to be answered.

One vital information authors leverage is that we can generate powerful adversarial examples by attacking shallow layers. I would encourage the authors to add appropriate references.
A vital piece of the technique is missing, like how is the optimization problem in Eq. 6 solved. I would encourage the authors to include such details in the main paper.

---

> ### Author Response · Authors · 2022-08-02
> **Response to Reviewer SEqv  (Part 4/4)**
>
>
> **References**
>
>
> [1] Adversarial Manipulation of Deep Representations. Sabour et al. ICLR 2016
>
> [2] Unsupervised Representation Learning by Predicting Image Rotations. Gidaris et al. ICLR 2018
>
> [3] Towards deep learning models resistant to adversarial attacks. Madry et al. ICLR 2018
>
> [4] Boosting adversarial attacks with momentum. Dong et al. CVPR 2018
>
> [5] Improving transferability of adversarial examples with input diversity. Xie et al. CVPR 2019
>
> [6] Black-box Adversarial Attacks with Limited Queries and Information. Ilyas et al. ICML 2018
>
> [7] Characterizing adversarial subspaces using local intrinsic dimensionality. Ma et al. In ICLR, 2018
>
> [8] A Simple Unified Framework for Detecting Out-of-Distribution Samples and Adversarial Attacks. Lee et al. NeurIPS 2018
>
> [9]  Adversarial examples in the physical world. Kurakin et al. ICLR 2016.
>
> [10]  Explaining and harnessing adversarial examples. Goodfellow  et al. ICLR  2015.
>
> [11] Very deep convolutional networks for large-scale image recognition. Simonyan et al. ICLR 2015.
>
> [12] Rethinking the inception architecture for computer vision.  Szegedy  et al. CVPR 2016.
>
> [13] Deep residual learning for image recognition. He et al. CVPR 2016.
>
> [14] Densely connected convolutional networks. Huang et al. CVPR 2017.
>
> [15] Squeeze-and-excitation networks. Hu  et al.  CVPR 2018.
>
> [16] Wide residual networks. Zagoruyko  et al.   BMVC  2016.
>
> [17] Mobilenetv2: Inverted residuals and linear bottlenecks. Sandler et al.  CVPR 2018.
>
> [18]  A simple framework for contrastive learning of visual representations. Chen et al. ICML 2020.
>
> [19] Reliable evaluation of adversarial robustness with an ensemble of diverse parameter-free attacks.  Croce  et al. ICML 2022.
>
> [20] ShuffleNet V2: Practical Guidelines for Efficient CNN Architecture Design.  Ma  et al.  ECCV  2018.

---

> > ### Comment · Reviewer_SEqv · 2022-08-06
> > **Response to Rebuttal**
> >
> > Dear Authors,
> >
> > Thanks for addressing/answering all my questions, and I am satisfied with the responses. I have updated my scores to reflect the same.

---

> > > ### Author Response · Authors · 2022-08-06
> > > **Thanks for raising the score.**
> > >
> > > Dear Reviewer SEqv,
> > >
> > > Glad to hear that your concerns/questions are addressed well. Thank you for raising the score.
> > >
> > > Best regards,
> > >
> > > Authors of #565

---

> ### Author Response · Authors · 2022-08-02
> **Response to Reviewer SEqv (Part 3/4)**
>
> > Q7: "How useful are the examples generated by ETF in improving the robustness of the models?"
>
> **A7:**
> Following your kind suggestion, we perform adversarial training using ETF to exploit whether examples generated by ETF can improve model robustness. Results are reported in [TABLE 2-4]. Unfortunately, adversarial training with examples generated by ETF attack cannot enhance robustness.
>
>
> **TABLE 2-4**: The robust accuracy of the normally trained model and the model trained with ETF adversarial examples.
>
> | Model                             | clean  | FGSM[10]   | PGD   | Autoattack[19]  |
> |-----------------------------------|--------|--------|-------|-------------|
> | Adversarial trained model  | 88.80% | 15.66% | 0.10% | 0.00%       |
> | Normal model                      | 94.02%    | 15.12% | 0.00% | 0.00%       |
>
>
> >Q8:"In addition, as the performance of the surrogate is tied to the samples used for training it, I would encourage the authors to run the experiments on different sample sets and report the variance."
>
> **A8:**
> Thanks for your valuable suggestion. We have reported the variance in the revision and conducted the experiments on the CIFAR10 dataset, see [TABLE 2-5]. The results demonstrate that ETF is relatively robust to a different set of samples.
>
> **TABLE 2-5:** The accuracy of 4 target models normally trained on the CIFAR10 dataset and evaluated on 1,000 adversarial examples generated by lightweight black-box attacks or existing black-box attacks, under $\epsilon \leq 0.1$. The Shallow-(PGD, MI, DI, TI) mean applying PGD, MI, DI and TI to the shallow layers of the model. Deep-(PGD, MI, DI and TI) mean applying PGD, MI, DI and TI to the model’s output. EFT-(PGD, MI, DI and TI) mean applying ETF combined with PGD, MI, DI or TI to the shallow layers. "Deep*" means the black-box setting where the surrogate models are trained on the training data the same as the seven target models.
> | Model         | VGG19[11]      | RN56[13]       | MobileNet[17]  | ShuffleNet[20]  | Avg        |
> |---------------|------------|------------|------------|-------------|------------|
> | clean         | 93.91%     | 94.37%      | 93.72%    | 92.98%       | 93.74%     |
> | Deep-PGD      | 59.45  $\pm$ 0.34 % | 57.58  $\pm$ 0.46 % | 45.21 $\pm$  0.27% | 52.32  $\pm$  0.37 % | 53.64   $\pm$ 0.78 %|
> | Deep-MI       | 53.44  $\pm$ 0.75 %  | 52.17  $\pm$ 0.65 % | 44.25  $\pm$ 0.34 % | 49.80  $\pm$  0.35 % | 49.91  $\pm$ 0.58  %|
> | Deep-DI       | 60.24  $\pm$ 0.19 % | 58.63  $\pm$ 0.34 % | 47.67  $\pm$ 0.31 % | 54.34  $\pm$  0.62 % | 55.22  $\pm$ 0.52 % |
> | Deep-TI       | 64.51  $\pm$ 0.38 % | 59.85  $\pm$ 0.60 % | 48.80  $\pm$ 0.59 % | 56.88  $\pm$  0.44 % | 57.51  $\pm$ 0.42 % |
> |  Shallow-PGD  | 27.17  $\pm$ 0.74 % | 31.06  $\pm$ 0.55 % | 22.83  $\pm$ 0.66 % | 28.14  $\pm$  0.76 % | 27.30  $\pm$  0.81 % |
> | Shallow-MI    | 32.43  $\pm$ 0.98 % | 36.42  $\pm$ 1.01 % | 31.84  $\pm$ 0.79 % | 30.76  $\pm$  0.94 % | 32.86 $\pm$  0.94 % |
> | Shallow-DI    | 25.65  $\pm$ 0.56 % | 30.27  $\pm$ 0.51 % | 22.61  $\pm$ 0.38 % | 27.22    $\pm$ 0.55 % | 26.43  $\pm$ 0.45 % |
> | Shallow-TI    | 28.66   $\pm$ 0.45 % | 31.35  $\pm$ 0.33 %  | 27.20  $\pm$ 0.44 % | 29.48   $\pm$  0.63 % | 29.17   $\pm$ 0.56 % |
> | ETF-PGD       | 21.27   $\pm$ 0.27 % | 25.85  $\pm$ 0.84 %  | **20.03** $\pm$ 0.65 % | 22.37   $\pm$  0.44 % | 22.38 $\pm$  0.53 %  |
> | ETF-MI        | **20.75** $\pm$ 0.55 % | **24.36** $\pm$ 0.35 % | 20.51  $\pm$ 0.34 % | **19.68** $\pm$ 0.23 % | **21.32** $\pm$ 0.42 % |
> | ETF-DI        | 21.37   $\pm$ 0.37 % | 26.46  $\pm$ 0.27 %  | 21.11   $\pm$ 0.69 % | 23.14    $\pm$ 0.36 %  | 23.02   $\pm$ 0.55 % |
> | ETF-TI        | 25.48   $\pm$ 0.41 % | 30.26  $\pm$ 0.23 %  | 23.37  $\pm$  0.51 % | 26.34   $\pm$ 0.25 %| 26.36  $\pm$ 0.39 %  |
> | Deep*-PGD     | 4.63    $\pm$ 0.54 % | 0.81   $\pm$ 0.74 %  | 3.79   $\pm$  0.28 % | 3.21    $\pm$ 0.32 %  | 3.11   $\pm$ 0.47 %  |
> | Deep*-MI      | 4.72    $\pm$ 0.20 % | 0.96   $\pm$ 0.36 %  | 4.36   $\pm$  0.12 % | 3.78    $\pm$ 0.25 %  | 3.45   $\pm$ 0.33 %  |
> | Deep*-DI      | 4.63    $\pm$ 0.17 % | 0.81   $\pm$ 0.67 %  | 2.38   $\pm$  0.53 % |  3.34   $\pm$ 0.43 %  | 2.79   $\pm$ 0.47 %  |
> | Deep*-TI      | 4.66    $\pm$ 0.18 % | 0.84   $\pm$ 0.25 %  | 3.78   $\pm$  0.46 % | 3.67    $\pm$ 0.31 %  | 3.23   $\pm$ 0.32 %  |
> | Auto-attack[19]   | 0.00%      | 0.00%      | 0.00%      | 0.00%       | 0.00%      |

---

> ### Author Response · Authors · 2022-08-02
> **Response to Reviewer SEqv (Part 2/4)**
>
> > Q3: "Does surrogate model architecture impact the success rate of the proposed technique?"
>
> **A3:**
> Thank you for your kind suggestions, and we add the following results and analysis in our revision, where we instantiate the shallow layers with different model architectures containing ResNet, VGG, and SENet. The results are reported in [TABLE 2-2], demonstrating that our EFT is powerful across various model architectures.
>
> TABLE 2-2: Model accuracy under ETF attack with different architectures, containing SENet, VGG11, and ResNet18.
> | Model         | VGG19[11]      | Inception_v3[12] | RN152[13]      | DenseNet161[14] | SENet[15]     | WRN[16]        | MobileNet[17]  | Avg        |
> |---------|------------|--------------|------------|-------------|-----------|------------|------------|------------|
> | clean         | 67.43%     | 64.35 %      | 74.21 %    | 73.34%      | 51.28%    | 73.22%     | 65.06%     | 66.99%     |
> | SENet[15]   | 23.44%     | 28.42%       | 35.07%     | 31.64%      | 6.73%     | 28.19%     | 11.80%     | 23.61%     |
> | VGG11[11]   | 18.20%     | 22.65%       | 27.24%     | 26.33%      | **6.47%** | 23.16%     | 12.69%     | 19.53%     |
> | ResNet18[13]  | **14.11%** | **20.22%**   | **24.20%** | **24.74%**  | 6.96%     | **20.73%** | **10.66%** | **17.37%** |
>
> > Q4: "Does the technique also work in other domains like NLP?"
>
> **A4**:
> Following much of previous works [3,4,5,6], we conduct experiments in the area of image classification. We also believe that it is an exciting problem to study the effectiveness of ETF in the field of NLP, but it remains challenging to use ETF for NLP. For instance, it is unclear in the NLP domain whether critical differences exist between those models learned from a few data and those learned from extensive training data, which is beyond the scope of this work. We sincerely appreciate your comment and will explore such an interesting problem in the future.
>
> > Q5: "The surrogate model is trained in a contrastive manner. Can other self-supervision tasks like rotation be used to train it?(ref. Unsupervised Representation Learning by Predicting Image Rotations)"
>
> **A5:**
> Thanks for your insightful question, we agree that exploring different strategies to train the shallow model is exciting for further improvement of the performance of lightweight black-box attacks, as shallow layers play an important role in lightweight black-box attacks. Thus, we generate adversarial examples using EFT with shallow layers trained with a rotation prediction task [2] and report the results in [TABLE 2-3]. We can see that shallow layers trained with the rotation prediction task is slightly worse than using the contrastive strategy, but the performance can also reduce the model accuracy significantly.
>
> **TABLE 2-3:**  The unsupervised representation Learning [2] applies to training the lightweight surrogate model.  The experiment is conducted on the ImageNet validation set.  "Classification" means the result of using cross-entropy loss to train lightweight surrogate models with label information to mount  ETF attacks.
>
> | Model         | VGG19[11]      | Inception_v3[12] | RN152[13]      | DenseNet161[14] | SENet[15]     | WRN[16]        | MobileNet[17]  | Avg        |
> |----------------|------------|--------------|------------|-------------|-----------|------------|------------|------------|
> | clean         | 67.43%     | 64.35 %      | 74.21 %    | 73.34%      | 51.28%    | 73.22%     | 65.06%     | 66.99%     |
> | SimCLR[18]    | 15.32%     | **18.54%**   | 25.81%     | 24.77%      | **6.64%** | 22.90%     | 11.34%     | 17.91%     |
> | Rotation[2]       | 19.07%     | 21.79%       | 27.30%     | 28.85%      | 7.66%     | 23.94%     | 12.51%     |  20.16%    |
> | Classification | **14.11%** | 20.22%       | **24.20%** | **24.74%**  | 6.96%     | **20.73%** | **10.66%** | **17.37%** |
>
> > Q6:"For table 1, were the surrogate model trained using labels or in a contrastive manner?"
>
> **A6:**
> Thanks for pointing out the potentially confusing problem, we have fixed it in the revision. All surrogate models except those used in Table 4 (in the paper) are trained via an instance discrimination task, i.e., using labels.

---

> ### Author Response · Authors · 2022-08-02
> **Response to Reviewer SEqv (Part 1/4)**
>
> **Response to Reviewer SEqv**
>
> We sincerely thank you for your constructive comments and positive feedback about our work! Please see our detailed responses to your comments and suggestions below.
>
> **Response to [Weakness]**
>
> > W1: "The paper is a bit hard to follow, and the writing needs improvement."
>
> **R1**: Sorry for our unclear description, we will definitely improve our writing and paper organization in our revision.
>
> > W2: "One vital information authors leverage is that we can generate powerful adversarial examples by attacking shallow layers. I would encourage the authors to add appropriate references."
>
> **R2:**
> Following your valuable suggestion, we have highlighted the reference in the revision to support the statement. According to the conclusion drawn from [1], adversarial examples can be generated by attacking shallow layers, i.e., perturbing representations at shallow layers of deep neural networks.
>
> > W3: "A vital piece of the technique is missing, like how is the optimization problem in Eq. 6 solved. I would encourage the authors to include such details in the main paper."
>
> **R3:**
> Thank you for pointing out the confusing problem. We have added the following description to the revision following your valuable suggestion. Given the strength of perturbations, we perform a min-max optimization to generate adversarial examples. Specifically, we solve the inner maximization problem by generating perturbations in the feature space, given a perturbed adversarial example. This step aims to mitigate the approximation error. The outer minimization problem is solved by finding adversarial perturbations in the input space, the same as the adversarial example generation. After iterative generation of perturbations, we can consider that adversarial examples are generated by attacking a model with a reduced approximation error.
>
>
> **Responses to [Questions]**
>
> > Q1: "How imperceptible are the examples generated by the proposed technique?"
>
> **A1:**
> Thank you for your kind suggestion, and we find that the resultant perturbations are truly imperceptible for our ETF. Please refer to Figure 2 (in the revision) for the visualization with deep*-PGD attack (using training images), deep-PGD attack (using test images), and lightweight black-box attack. We have added the figure and analysis to the revision.
>
> > Q2:"How good are examples generated by the proposed technique in evading the recent class of adversarial example detection methods?"
>
> **A2:**
> Thanks for your constructive suggestion. We employ a recent detection method [7,8] to detect adversarial examples generated by different attack methods, e.g., FGSM, PGD, BIM, and ETF. All settings are the same as that used in the paper, and the results are reported in [TABLE 2-1]. We can see that ETF performs better than the baselines, i.e., having a high probability of evading detection methods.
>
> TABLE 2-1: Performance of adversarial detection against four attacks, metric to evaluate the detection performance can be found in [7,8].
>
> |  Mahalanobis[8]   |           |           |           |           |           |
> |---------------|-----------|-----------|-----------|-----------|-----------|
> | Method        | TNR       | AUROC     | DTACC     | AUIN      | AUOUT     |
> | BIM[9]           | 99.99%     | 99.99%     | 99.86%     | 99.86%     | 99.71%     |
> | FGSM[10]          | 98.89%     | 99.88%     | 98.89%     | 99.66%     | 99.24%     |
> | Deep*-PGD     | 97.22%     | 99.58%     | 97.92%     | 99.64%     | 99.05%     |
> | ETF           | **96.67%** | **98.73%** | **96.94%** | **98.75%** | **97.98%** |
>
> |   LID[7]  |           |           |           |           |           |
> |-----------|-----------|-----------|-----------|-----------|-----------|
> | Method    | TNR       | AUROC     | DTACC     | AUIN      | AUOUT     |
> | BIM[9]       | 99.99%     | **98.81%** | 98.33%     | 99.77%     | 99.33%     |
> | FGSM[10]      | 99.99%     | 99.99%     | 99.99%     | 99.72%     | 99.44%     |
> | Deep*-PGD | 99.99%     | 99.99%     | 99.99%     | 99.86%     | 99.72%     |
> | ETF       | **97.78%** | 99.58%     | **97.22%** | **99.51%** | **98.68%** |

---

### Official Review · Reviewer_LA1D · 2022-07-12

**Rating:** 6
**Confidence:** 4
**Soundness:** 3 good
**Presentation:** 3 good
**Contribution:** 2 fair

**Summary:**

The paper looks at the interesting task of no-box attacks. This is a setting that is even harder than black-box attacks, where typically the attacker has access to either the training data and can construct a surrogate model or the model outputs to estimate gradients. In the no-box setting considered in this paper, the attacker only has access to a small number of samples (1k on ImageNet) that are correctly classified by the target model. The attack is based on feature space perturbations, as only the shallow layers of DNNs can be approximated with this little data and the authors propose the Error TransFormer to alleviate issues caused by approximations errors.

They first train a Shallow DNN on the limited data pool. As the shallow layers do not yield a label output, instead of the standard adversarial objective used for FGSM/PGD attacks, their attack objective transforms the feature representation of an image into that of a given image that comes from a different class. To mitigate the difference between the target model and the surrogate model, they employ a min-max strategy that combines feature and parameter space.
They evaluate their approach on various ImageNet source models.



**Questions:**

- The paper only considers L-inf perturbations. While they are the most common, does the method work for L0, L1 and L2 too?

- In 4.1 you say that the model has access to label information on the limited data pool. From the insufficient supervision information ablation, I also take that this is used during training. However 4.1 also motivates that for shallow attacks we only need to train shallow nets with an unlabeled contrastive loss. Which loss (Eq (1) vs (2)) is used for the actual training of the surrogate model?

- You use heavy data augmentation, what is the impact of that and did you experiment with different ones?

**Limitations:**

Obviously, attack methods can in general be used in a malicious way, especially if we go to more realistic scenarios. I think adding a sentence about that in C would make sense.

**Strengths And Weaknesses:**

Strengths:

- The paper is well written and easy to understand.

- The setting is very interesting. Assumptions from the standard black-box setting are often not realistic and the no-box setting is more applicable in real world scenarios.

- Their method is simple and easy to implement.

- The evaluation is clear. Deep models can not be trained in the no-box setting, shallow models work much better but can be improved by their ETF approach.

Weakenesses:

- The novelty is somewhat limited. A similar min-max objective similar to ETF in weight space has for example been explored already in "Adversarial Weight Perturbation Helps Robust Generalization" by Wu et al. Most other ideas like surrogate training of a shallow model also appeared in previous papers.

- The evaluation focuses on Imagenet only. While this is certainly an interesting setting, it might be worth to also add some experiments on other datasets. For example, while CIFAR10 is a much easier dataset, there also exist much more robust models and it might be interesting to evaluate the performance of the method to the most robust models for example from RobustBench.


- In Table 1-3, their should be another row for the ground truth robust accuracy in a white-box setting. While this is not computable exactly, AutoAttack should yield a fairly accurate estimate.

- Table 1 is somewhat hard to read. I understand that boldfacing the single best result per column is not optimal as it does not make sense to compare different attacks, but it would be easier to understand the general message of the table, which clearly is that ETF is the best method in the no-box setting, by some form of highlighting.

---

> ### Author Response · Authors · 2022-08-02
> **Response to Reviewer LA1D (Part 4/4)**
>
>
> **References**
>
> [1] Adversarial Weight Perturbation Helps Robust Generalization. Wu et al. NeurIPS 2020.
>
> [2] A Simple Framework for Contrastive Learning of Visual Representations. Chen et al. ICML 2020.
>
> [3] A critical analysis of self-supervision, or what we can learn from a single image. Asano et al. ICLR 2022.
>
> [4] RobustBench: a standardized adversarial robustness benchmark. Croce et al. NeurIPS 2021.
>
> [5] Reliable Evaluation of Adversarial Robustness with an Ensemble of Diverse Parameter-free Attacks. Croce et al. ICML 2020.
>
> [6] EAD: Elastic-Net Attacks to Deep Neural Networks via Adversarial Examples. Chen et al. AAAI 2018.
>
> [7] One Pixel Attack for Fooling Deep Neural Networks. Su et al. IEEE Trans on Evolutionary Computation, 2019.
>
> [8] Improving transferability of adversarial examples with input diversity. Xie et al. CVPR 2019.
>
> [9] Evading Defenses to Transferable Adversarial Examples by Translation-Invariant Attacks. Dong et al. CVPR 2019.
>
> [10] Black-box Adversarial Attacks with Limited Queries and Information. Ilyas et al. ICML 2018.
>
>
> [11] Very deep convolutional networks for large-scale image recognition. Simonyan et al. ICLR 2015.
>
> [12] Rethinking the inception architecture for computer vision.  Szegedy  et al. CVPR 2016.
>
> [13] Deep residual learning for image recognition. He et al. CVPR 2016.
>
> [14] Densely connected convolutional networks. Huang et al. CVPR 2017.
>
> [15] Squeeze-and-excitation networks. Hu  et al.  CVPR 2018.
>
> [16] Wide residual networks. Zagoruyko  et al.   BMVC  2016.
>
> [17]Mobilenetv2: Inverted residuals and linear bottlenecks. Sandler et al.  CVPR 2018.
>
> [18] ShuffleNet V2: Practical Guidelines for Efficient CNN Architecture Design.  Ma  et al.  ECCV  2018.
>
> [19] Improving Robustness using Generated Data. Gowal  et al. NeurIPS 2021.
>
> [20] Stable neural ode with lyapunov-stable equilibrium points for defending against adversarial attacks.       Kang et al. NeurIPS 2021.
>
> [21] Robustness and Accuracy Could Be Reconcilable by (Proper) Definition. Pang et al.  ICML 2022.
>
> [22] Robust learning meets generative models: Can proxy distributions improve adversarial robustness? Sehwag et al. ICLR 2022.
>
> [23] Reliable evaluation of adversarial robustness with an ensemble of diverse parameter-free attacks.  Croce  et al. ICML 2022.
>
> [24] Towards deep learning models resistant to adversarial attacks. Madry  et al. ICLR 2018.
>
> [25] Boosting adversarial attacks with momentum. Dong  et al. CVPR 2018.
>
> [26] Improving transferability of adversarial examples with input diversity. Xie et al. CVPR 2019.

---

> ### Author Response · Authors · 2022-08-02
> **Response to Reviewer LA1D (Part 3/4)**
>
>
> > W3:  "In Table 1-3, their should be another row for the ground truth robust accuracy in a white-box setting."
>
> **R3:**
> Thanks for your valuable suggestion. To make the table clear, we add one raw in all mentioned tables to show the white-box attack performance using AutoAttack [5]. In the revision, we have replaced tables in the old version with the new ones.
>
> > W4: "It would be easier to understand the general message of the table, which clearly is that ETF is the best method in the no-box setting, by some form of highlighting."
>
> **R4:**
> Thank you for pointing out the potentially confusing problem. We have highlighted ETF in tables and replaced the confusing tables with new ones.
>
> **Response to [Questions]**
>
> > Q1: "The paper only considers L-inf perturbations. While they are the most common, does the method work for L0, L1 and L2 too?"
>
> **A1:**
> We mainly conduct experiments with L-inf perturbation since it is widely adopted in many previous works [8,9,10]. To further demonstrate the power of our ETF, we follow your kind suggestion in testing $\ell_2$-norm perturbation as an example. The results are reported in [TABLE 1-4], which further demonstrate the effectiveness of our proposal. Considering that $\ell_1$ and $\ell_0$ perturbations require careful design [6,7], it is beyond the scope of our work, so we leave it as our future work.
>
> **TABLE 1-4**: The classification accuracy evaluation on **$\ell_2$-norm** attacks. The experiment is conducted on the ImageNet validation. Following the previous work[25] about **$\ell_2$-norm** attacks, the maximum disturbance $\varepsilon$ is set to 16 $\sqrt[2]{N}$ where N is the dimension of input to attacks.
>
> | Model         | VGG19[11]      | Inception_v3[12] | RN152[13]      | DenseNet161[14] | SENet[15]     | WRN[16]        | MobileNet[17]  | Avg        |
> |----------------|------------|--------------|------------|-------------|-----------|------------|------------|------------|
> | clean         | 67.43%     | 64.35 %      | 74.21 %    | 73.34%      | 51.28%    | 73.22%     | 65.06%     | 66.99%     |
> | Deep-PGD    | 37.73% | 42.75% | 51.04% | 51.96% | 17.48% | 50.61% | 31.07% | 40.38% |
> | Shallow-PGD | 25.73% | 31.51% | 44.96% | 43.72% | 8.58%  | 40.62% | 18.73% | 30.55% |
> | ETF-PGD     | 22.16% | 27.03% | 34.87% | 37.94% | 11.28% | 29.63% | 16.17% | 25.58% |
> | Deep*-PGD   | 7.65%  | 22.88% | 11.44% | 11.23% | 4.56%  | 9.69%  | 8.03%  | 10.78% |
> | Autoattack[23]     | 0.00%      | 0.00%        | 0.00%      | 0.00%       | 0.00%     | 0.00%      | 0.00%      | 0.00%      |
>
>
>
> >Q2: "Which loss (Eq (1) vs (2)) is used for the actual training of the surrogate model?"
>
> **A2:**
> Thank you for pointing out the potentially confusing problem. Eq. (1) is used in most experiments, i.e., Table 1, 2, and 3 (in the paper), as label information is usually available. Eq. (2) is a promising candidate, especially for the scenarios where the adversary cannot access the label information. Thus, we also report the results in Table 4 (in the paper, termed as Unsupervised) to show that we can generate powerful adversarial examples in the no-box threat model, even if the label information is unavailable. We have added the above clarification into the revision.
>
> >Q3: "You use heavy data augmentation, what is the impact of that and did you experiment with different ones?"
>
> **A3:**
> We follow the empirical conclusion suggested in [3], where heavy data augmentation is vital for training appropriate shallow models. Because appropriate shallow models are necessary for mounting lightweight black-box attacks, data augmentation plays a crucial role and is heavily used in our experiments. This is supported by results shown in [TABLE 1-5], where we report the performance of lightweight black-box attacks with and without data augmentation. The results and conclusion have been added to the revision.
>
>
> **TABLE 1-5**：The impact of augmentation to ETF attacks. "No-Aug" means the effect of the attack on the ETF using the surrogate model without augmentation for training. This experiment is conducted on the ImageNet validation. The best results are in bold.
>
> | Model         | VGG19[11]      | Inception_v3[12] | RN152[13]      | DenseNet161[14] | SENet[15]     | WRN[16]        | MobileNet[17]  | Avg        |
> |--------|------------|--------------|------------|-------------|-----------|------------|------------|------------|
> | clean         | 67.43%     | 64.35 %      | 74.21 %    | 73.34%      | 51.28%    | 73.22%     | 65.06%     | 66.99%     |
> | No-Aug    | 34.58%     | 39.17%       | 46.25%     | 50.06%      | 10.42%    | 45.10%     | 22.92%     | 35.50%     |
> | Aug | **14.11%** | **20.22%**   | **24.20%** | **24.74%**  | **6.96%** | **20.73%** | **10.66%** | **17.37%** |

---

> ### Author Response · Authors · 2022-08-02
> **Response to Reviewer LA1D (Part 2/4)**
>
> > W2: "It might be interesting to evaluate the performance of the method using CIFAR10 and the most robust models for example from CIFAR10 RobustBench."
>
> **R2:**
> We conducted extensive experiments in Section 5 and Appendix B, and the results verified the power of our ETF. However, your kind suggestion is also great for further solidifying our evaluation. Thus, we conduct the experiments on the CIFAR10 dataset, see [TABLE 1-2], and evaluate the robustness of models downloaded from RobustBench [4], see [TABLE 1-3]. The conclusion drawn from [TABLE 1-2] and [TABLE 1-3] is consistent with that drawn from Table 1 (in the paper) evaluating on ImageNet dataset.
>
> **TABLE 1-2**: Evaluate the performances of different attacks  on  CIFAR10. Here, experiments of "Deep-, Shallow-, ETF-" are conducted in the no-box threat model. "Deep*" means the black-box setting where the surrogate models are trained on the training data the same as the seven target models. "PGD[24], MI[25], DI[26], TI[27]" is applied to the different settings and methods. Auto-attack[23] is used for testing the robustness of the target models, so it adopts the white-box setting to mount the seven target model. $\varepsilon \leq 0.1$ in $\ell_\infty$-norm.
>
> | Model         | VGG19[11]      | RN56[13]       | MobileNet[17]  | ShuffleNet[18]  | Avg        |
> |---------------|------------|------------|------------|-------------|------------|
> | clean         | 93.91%     | 94.37%      | 93.72%    | 92.98%       | 93.74%     |
> | Deep-PGD      | 59.45  $\pm$ 0.34 % | 57.58  $\pm$ 0.46 % | 45.21 $\pm$  0.27% | 52.32  $\pm$  0.37 % | 53.64   $\pm$ 0.78 %|
> | Deep-MI       | 53.44  $\pm$ 0.75 %  | 52.17  $\pm$ 0.65 % | 44.25  $\pm$ 0.34 % | 49.80  $\pm$  0.35 % | 49.91  $\pm$ 0.58  %|
> | Deep-DI       | 60.24  $\pm$ 0.19 % | 58.63  $\pm$ 0.34 % | 47.67  $\pm$ 0.31 % | 54.34  $\pm$  0.62 % | 55.22  $\pm$ 0.52 % |
> | Deep-TI       | 64.51  $\pm$ 0.38 % | 59.85  $\pm$ 0.60 % | 48.80  $\pm$ 0.59 % | 56.88  $\pm$  0.44 % | 57.51  $\pm$ 0.42 % |
> |  Shallow-PGD  | 27.17  $\pm$ 0.74 % | 31.06  $\pm$ 0.55 % | 22.83  $\pm$ 0.66 % | 28.14  $\pm$  0.76 % | 27.30  $\pm$  0.81% |
> | Shallow-MI    | 32.43  $\pm$ 0.98 % | 36.42  $\pm$ 1.01 % | 31.84  $\pm$ 0.79 % | 30.76  $\pm$  0.94 % | 32.86 $\pm$  0.94 % |
> | Shallow-DI    | 25.65  $\pm$ 0.56 % | 30.27  $\pm$ 0.51 % | 22.61  $\pm$ 0.38 % | 27.22    $\pm$ 0.55 % | 26.43  $\pm$ 0.45 % |
> | Shallow-TI    | 28.66   $\pm$ 0.45 % | 31.35  $\pm$ 0.33 %  | 27.20  $\pm$ 0.44 % | 29.48   $\pm$  0.63 % | 29.17   $\pm$ 0.56 % |
> | ETF-PGD       | 21.27   $\pm$ 0.27 % | 25.85  $\pm$ 0.84 %  | **20.03** $\pm$ 0.65 % | 22.37   $\pm$  0.44 % | 22.38 $\pm$  0.53 %  |
> | ETF-MI        | **20.75** $\pm$ 0.55 % | **24.36** $\pm$ 0.35 % | 20.51  $\pm$ 0.34 % | **19.68** $\pm$ 0.23 % | **21.32** $\pm$ 0.42 % |
> | ETF-DI        | 21.37   $\pm$ 0.37 % | 26.46  $\pm$ 0.27 %  | 21.11   $\pm$ 0.69 % | 23.14    $\pm$ 0.36 %  | 23.02   $\pm$ 0.55 % |
> | ETF-TI        | 25.48   $\pm$ 0.41 % | 30.26  $\pm$ 0.23 %  | 23.37  $\pm$  0.51 % | 26.34   $\pm$ 0.25 %| 26.36  $\pm$ 0.39 %  |
> | Deep*-PGD     | 4.63    $\pm$ 0.54 % | 0.81   $\pm$ 0.74 %  | 3.79   $\pm$  0.28 % | 3.21    $\pm$ 0.32 %  | 3.11   $\pm$ 0.47 %  |
> | Deep*-MI      | 4.72    $\pm$ 0.20 % | 0.96   $\pm$ 0.36 %  | 4.36   $\pm$  0.12 % | 3.78    $\pm$ 0.25 %  | 3.45   $\pm$ 0.33 %  |
> | Deep*-DI      | 4.63    $\pm$ 0.17 % | 0.81   $\pm$ 0.67 %  | 2.38   $\pm$  0.53 % |  3.34   $\pm$ 0.43 %  | 2.79   $\pm$ 0.47 %  |
> | Deep*-TI      | 4.66    $\pm$ 0.18 % | 0.84   $\pm$ 0.25 %  | 3.78   $\pm$  0.46 % | 3.67    $\pm$ 0.31 %  | 3.23   $\pm$ 0.32 %  |
> | Auto-attack[23]   | 0.00%      | 0.00%      | 0.00%     | 0.00%        | 0.00%     |
>
>
> **TABLE 1-3**: The attacks on the most robust models from CIFAR10 RobustBench. The robustness model is trained by the different adversarial defense method,$\varepsilon \leq 0.1$ in $\ell_\infty$-norm.
>
> |  Setting  |    Model    | Gowal2021[19]  | Kang2021[20]  | Pang2022[21]  | Sehwag2021[22]  |    Avg   |
> |:---------:|:-----------:|:----------:|:---------:|:---------:|:-----------:|:--------:|
> |           |    clean    |   89.99%   |   92.50%  |   87.56%  |    86.55%   |   89.15% |
> |   No-box  |  ETF-PGD    |   72.01%   |   72.86%  |   72.50%  |    67.44%   |   71.20% |
> | Black-box |  Deep*-PGD  |   83.53%   |   88.06%  |   83.17%  |    79.44%   |   83.55% |
> | White-box | Auto-attack[23] |    8.05%   |   21.13%  |   7.46%   |    6.53%    |   10.79% |

---

> ### Author Response · Authors · 2022-08-02
> **Response to Reviewer LA1D (Part 1/4)**
>
>  **Response to Reviewer LA1D:**
>
> We sincerely thank for your constructive comments and positive feedback about our work! Please see our detailed responses to your comments and suggestions below.
>
> **Response to [Weakness]**
>
> > **W1**:  "The novelty is somewhat limited: a) min-max objective similar to ETF in weight space has for example been explored; b) surrogate training of a shallow model also appeared in previous papers."
>
> **R1**:
> Thanks for your valuable comments. We have added explanations to the revision to highlight our contribution and the difference between our work and previous works [1,2].
>
> a) The min-max strategy is proposed to flatten the loss landscape in the weight space [1], but our work proposes performing feature space min-max optimization for approximation error minimization. We conduct experiments using the weight space min-max optimization to highlight the difference further in [TABLE 1-1].
>
> The results in Table [TABLE 1-1] demonstrate the superiority of feature space optimization. We suspect that the performance gain results from the fact that performing the min-max strategy in the feature space is more appropriate than the weight space optimization for the no-box threat model. This is because we know which perturbations are preferred in the feature space, e.g., towards features of guide images, but we have no idea about which perturbations are preferred in the weight space, i.e., no "guide models".
>
> b) Surrogate training of a shallow model is widely used in self-supervised learning [2], and recent work [3] demonstrates that models learned from a few images can approximate the shallow layers of models trained on millions of images. However, these works did not focus on adversarial attacks and did not explore an appropriate strategy for crafting compelling adversarial examples.
>
> **TABLE 1-1**:This experiment is conducted in 1000 samples randomly selected from ImageNet validation and evaluates the attack performance on seven different pre-trained models(loaded by torchvision). The structure of the lightweight model is Resnet18[13]. Feature_space refers to our ETF method, and Weight_space is achieved by the min-max strategy in the weight space[1]. $\varepsilon \leq 0.1$, $\ell_\infty$-norm. The best results are in bold.
>
> | Model         | VGG19[11]      | Inception_v3[12] | RN152[13]      | DenseNet161[14] | SENet[15]     | WRN[16]        | MobileNet[17]  | Avg        |
> |---------------|------------|--------------|------------|-------------|-----------|------------|------------|------------|
> | clean         | 67.43%     | 64.35 %      | 74.21 %    | 73.34%      | 51.28%    | 73.22%     | 65.06%     | 66.99%     |
> |  Weight_space | 29.43%     | 32.44%       | 40.11%     | 41.88%      | 10.12%    | 35.41%     | 19.27%     | 29.81%     |
> | Feature_space | **14.11%** | **20.22%**   | **24.20%** | **24.74%**  | **6.96%** | **20.73%** | **10.66%** | **17.37%** |

---

> ### Author Response · Authors · 2022-08-09
> **Looking forward to your reply.**
>
> Dear Reviewer LA1D:
>
> Thank you again for your valuable comments and constructive suggestions on our work. Would you mind checking the response to confirm whether it addresses your questions/concerns?
>
> Since we list many points in the response and the window for discussion is closing, we want to summarize our response here quickly. We hope this can help you quickly go through all feedback.
>
> In addition to clarifying the approach and experiment details, following your kind suggestions, we have added five experiments in respect to your constructive suggestions. Specifically:
>  - highlight the difference between the feature and weight space min-max strategy;
>  - verify the effectiveness using CIFAR10;
>  - test the robustness of ETF on RobustBench;
>  - evaluate the approach with $\ell_2$-norm perturbation;
>  - study the impact of data augmentation.
>
> All experiments have been added to the revision. Besides, we have revised the paper to improve the clarity and readability following your valuable suggestions.
>
> We'd be glad to answer any outstanding questions and look forward to any further discussions.
>
> Best regards,
>
> Authors of #565

---

### Review · Ethics_Reviewer_24hp · 2022-07-23

**Recommendation:**

The social impact section in Appendix C needs to be referenced in some form in the main paper body, or there should be an explicit section of the paper that addresses misuse.  Ideally the social impact section can be fleshed out more and encourage defense method development.

**Ethical Issues:**

Yes

**Ethics Review:**

Largely misuse concerns that are addressed in Appendix C

---

### Review · Ethics_Reviewer_ruS3 · 2022-07-27

**Recommendation:**

The concern could be addressed by adding the noted considerations to the main paper, and expanding to briefly discuss how more advanced black-box attacks could be misused.

**Ethical Issues:**

Yes

**Ethics Review:**

The paper targets improved black-box attacks on neural networks, which could be used to attack deployed models in safety-critical settings.

---

### Author Response · Authors · 2022-08-08
**Introduction to the revised version.**

Dear reviewers,

We have revised our paper following the insightful suggestions/comments from all the reviewers. The revision is in blue color:

**For the related works** We added explanations to claim the differences between ours and existing works, especially the min-max strategy.

**Further description of approach details** Following the reviewers' suggestion in the method details section, we made changes in Sec. 4 regarding the use of formulae, notation of symbols, and clarification of misunderstandings.

**Addition of experimental details** We add more descriptions related to the experimental details in Sec. 5.

**Visualization of Adversarial Examples** We add adversarial examples generated by our method in Sec 5.3.

We provide a link of our codes for reproducing the results: https://anonymous.4open.science/r/Error_TransFormer-7495/README.md

Best regards,

Authors of #565

---

### Meta-Review · Area_Chair_aPbW · 2022-08-31

**Recommendation:** Accept
**Confidence:** Certain

**Metareview:**

The paper presents a new method for generating black-box attacks with very limited data, i.e., the no-box case. The attack is based on feature transformations and the paper proposes error transformers (ETF) to alleviate issues with approximation errors. The reviewers believe the paper is technically solid and raised issues mainly to do with clarity and experiments. The authors provided a rebuttal and updated paper that thoroughly addressed those issues. All the reviewers raised their scores (including one that said to do so but did not update it). A good contribution to the field of adversarial learning.


**Award:**

No

---

### Decision · Program_Chairs · 2022-09-14

Accept